# PACKED-ENSEMBLES FOR EFFICIENT UNCERTAINTY ESTIMATION

**Olivier Laurent,**[1,2,*] **Adrien Lafage,**[2,*] **Enzo Tartaglione,**[3] **Geoffrey Daniel,**[1]
**Jean-Marc Martinez**,[1] **Andrei Bursuc**[4] & **Gianni Franchi**[2,†]

Université Paris-Saclay, CEA, SGLS,[1] U2IS, ENSTA Paris, Institut Polytechnique de Paris,[2]
LTCI, Télécom Paris, Institut Polytechnique de Paris,[3] valeo.ai[4]

## ABSTRACT

Deep Ensembles (DE) are a prominent approach for achieving excellent performance on key metrics such as accuracy, calibration, uncertainty estimation, and out-of-distribution detection. However, hardware limitations of real-world systems constrain to smaller ensembles and lower-capacity networks, significantly deteriorating their performance and properties. We introduce Packed-Ensembles (PE), a strategy to design and train lightweight structured ensembles by carefully modulating the dimension of their encoding space. We leverage grouped convolutions to parallelize the ensemble into a single shared backbone and forward pass to improve training and inference speeds. PE is designed to operate within the memory limits of a standard neural network. Our extensive research indicates that PE accurately preserves the properties of DE, such as diversity, and performs equally well in terms of accuracy, calibration, out-of-distribution detection, and robustness to distribution shift. We make our code available at github.com/ENSTA-U2IS/torch-uncertainty.

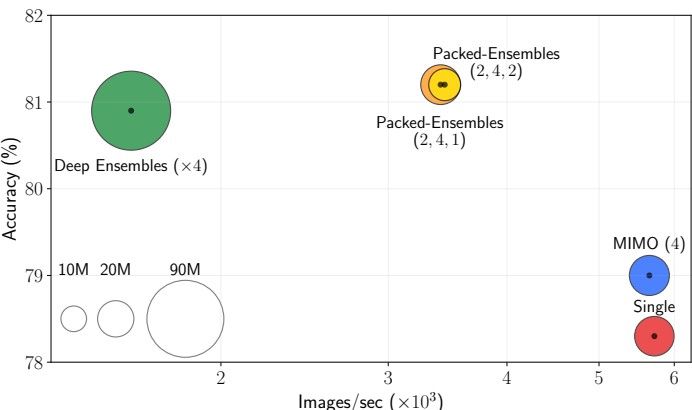

Figure 1: **Evaluation of computation cost *vs.* performance trade-offs for multiple uncertainty quantification techniques on CIFAR-100.** The y-axis and x-axis respectively show the accuracy and inference time in images per second. The circle area is proportional to the number of parameters. Optimal approaches are closer to the top-right corner. Packed-Ensembles strikes a good balance between predictive performance and speed.

## 1 INTRODUCTION

Real-world safety-critical machine learning decision systems such as autonomous driving (Levinson et al., 2011; McAllister et al., 2017) impose exceptionally high reliability and performance requirements across a broad range of metrics: accuracy, calibration, robustness to distribution shifts, uncertainty estimation, and computational efficiency under limited hardware resources. Despite significant improvements in performance in recent years, vanilla Deep Neural Networks (DNNs) still

---

*equal contribution      † corresponding author - gianni.franchi@ensta-paris.fr

exhibit several shortcomings, notably overconfidence in both correct and wrong predictions (Nguyen et al., 2015; Guo et al., 2017; Hein et al., 2019). Deep Ensembles (Lakshminarayanan et al., 2017) have emerged as a prominent approach to address these challenges by leveraging predictions from multiple high-capacity neural networks. By averaging predictions or by voting, DE achieves high accuracy and robustness since potentially unreliable predictions are exposed via the disagreement between individuals. Thanks to the simplicity and effectiveness of the ensembling strategy (Dieterich, 2000), DE have become widely used and dominate performance across various benchmarks (Ovadia et al., 2019; Gustafsson et al., 2020).

DE meet most of the real-world application requirements except computational efficiency. Specifically, DE are computationally demanding in terms of memory storage, number of operations, and inference time during both training and testing, as their costs grow linearly with the number of individuals. Their computational costs are, therefore, prohibitive under tight hardware constraints.

This limitation of DE has inspired numerous approaches proposing computationally efficient alternatives: multi-head networks (Lee et al., 2015; Chen & Shrivastava, 2020), ensemble-imitating layers (Wen et al., 2019; Havasi et al., 2020; Ramé et al., 2021), multiple forwards on different weight subsets of the same network (Gal & Ghahramani, 2016; Durasov et al., 2021), ensembles of smaller networks (Kondratyuk et al., 2020; Lobacheva et al., 2020), computing ensembles from a single training run (Huang et al., 2017; Garipov et al., 2018), and efficient Bayesian Neural Networks (Maddox et al., 2019; Franchi et al., 2020). These approaches typically improve storage usage, train cost, or inference time at the cost of lower accuracy and diversity in the predictions.

An essential property of ensembles to improve predictive uncertainty estimation is related to the diversity of its predictions. Perrone & Cooper (1992) show that the independence of individuals is critical to the success of ensembling. Fort et al. (2019) argue that the diversity of DE, due to randomness from weight initialization, data augmentation and batching, and stochastic gradient updates, is superior to other efficient ensembling alternatives, despite their predictive performance boosts. Few approaches manage to mirror this property of DE in a computationally efficient manner close to a single DNN (in terms of memory usage, number of forward passes, and image throughput).

In this work, we aim to design a DNN architecture that closely mimics properties of ensembles, in particular, having a set of independent networks, in a computationally efficient manner. Previous works propose ensembles composed of small models (Kondratyuk et al., 2020; Lobacheva et al., 2020) and achieve performances comparable to a single large model. We build upon this idea and devise a strategy based on small networks trying to match the performance of an ensemble of large networks. To this end, we leverage *grouped convolutions* (Krizhevsky et al., 2012) to delineate multiple subnetworks within the same network. The parameters of each subnetwork are not shared across subnetworks, leading to independent smaller models. This method enables fast training and inference times while predictive uncertainty quantification is close to DE (Figure 1).

In summary, our contributions are the following:

- We propose *Packed-Ensembles* (PE), an efficient ensembling architecture relying on grouped convolutions, as a formalization of structured sparsity for Deep Ensembles;
- We extensively evaluate PE regarding accuracy, calibration, OOD detection, and distribution shift on classification and regression tasks. We show that PE achieves state-of-the-art predictive uncertainty quantification.
- We thoroughly study and discuss the properties of PE (diversity, sparsity, stability, behavior of subnetworks) and release our PyTorch implementation.

## 2 BACKGROUND

In this section, we present the formalism for this work and offer a brief background on grouped convolutions and ensembles of DNNs. Appendix A summarizes the main notations in Table 3.

### 2.1 BACKGROUND ON CONVOLUTIONS

The convolutional layer (LeCun et al., 1989) consists of a series of cross-correlations between feature maps $\mathbf{h}^j \in \mathbb{R}^{C_j \times H_j \times W_j}$ regrouped in batches of size $B$ and a weight tensor $\boldsymbol{\omega}^j \in \mathbb{R}^{C_{j+1} \times C_j \times s_j^2}$

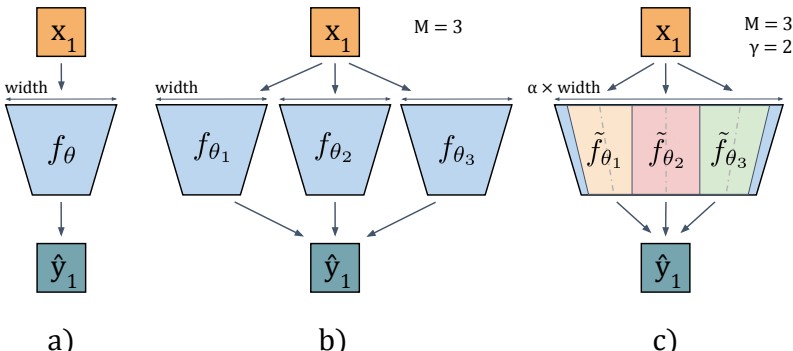

Figure 2: **Overview of the considered architectures:** (*left*) baseline vanilla network; (*center*) Deep Ensembles; (*right*) Packed-Ensembles-$(\alpha, M = 3, \gamma = 2)$.

with $C_j, H_j, W_j$ three integers representing the number of channels, the height and the width of $\mathbf{h}^j$ respectively. $C_{j+1}$ and $s_j$ are also two integers corresponding to the number of channels of $\mathbf{h}^{j+1}$ (the output of the layer) and the kernel size. Finally, $j$ is the layer's index and will be fixed in the following formulae. The bias of convolution layers will be omitted in the following for simplicity. Hence the output value of the convolution layer, denoted $\circledast$, is:

$$\mathbf{z}^{j+1}(c,:,:) = (\mathbf{h}^j \circledast \boldsymbol{\omega}^j)(c,:,:) = \sum_{k=0}^{C_j-1} \boldsymbol{\omega}^j(c, k, :, :) \star \mathbf{h}^j(k, :, :), \tag{1}$$

where $c \in [\![0, C_{j+1} - 1]\!]$ is the index of the considered channel of the output feature map, $\star$ is the classical 2D cross-correlation operator, and $\mathbf{z}^j$ is the pre-activation feature map such that $\mathbf{h}^j = \phi(\mathbf{z}^j)$ with $\phi$ an activation function.

To embed an ensemble of subnetworks, we leverage grouped convolutions, already used in ResNeXt (Xie et al., 2017) to train several DNN branches in parallel. The grouped convolution operation with $\gamma$ groups and weights $\boldsymbol{\omega}_\gamma^i \in \mathbb{R}^{C_{j+1} \times \frac{C_j}{\gamma} \times s_j^2}$ is given in (2), $\gamma$ dividing $C_j$ for all layers. Any output channel $c$ is produced by a specific group (set of filters), identified by the integer $\left\lfloor \frac{\gamma c}{C_{j+1}} \right\rfloor$, which only uses $\frac{1}{\gamma}$ of the input channels:

$$\mathbf{z}^{j+1}(c,:,:) = (\mathbf{h}^j \circledast \boldsymbol{\omega}_\gamma^j)(c,:,:)$$
$$= \sum_{k=0}^{\frac{C_j}{\gamma}-1} \boldsymbol{\omega}_\gamma^j(c, k, :, :) \star \mathbf{h}^j \left( k + \left\lfloor \frac{\gamma c}{C_{j+1}} \right\rfloor \frac{C_j}{\gamma}, :, : \right). \tag{2}$$

The grouped convolution layer is mathematically equivalent to a classical convolution where the weights are multiplied element-wise by the binary tensor $\text{mask}_m \in \{0, 1\}^{C_{j+1} \times C_j \times s_j^2}$ such that $\text{mask}_m^j(k, l, :, :) = 1$ if $\left\lfloor \frac{\gamma l}{C_j} \right\rfloor = \left\lfloor \frac{\gamma k}{C_{j+1}} \right\rfloor = m$ for each group $m \in [\![0, \gamma - 1]\!]$. The complete layer mask is finally defined as $\text{mask}^j = \sum_{m=0}^{\gamma-1} \text{mask}_m^j$ and the grouped convolution can therefore be rewritten as $\mathbf{z}^{j+1} = \mathbf{h}^j \circledast (\boldsymbol{\omega}^j \circ \text{mask}^j)$, where $\circ$ is the Hadamard product.

## 2.2 BACKGROUND ON DEEP ENSEMBLES

For an image classification problem, let us define a dataset $\mathcal{D} = \{\mathbf{x}_i, \mathbf{y}_i\}_{i=1}^{|\mathcal{D}|}$ containing $|\mathcal{D}|$ pairs of samples $\mathbf{x}_i = \mathbf{h}_i^0 \in \mathbb{R}^{C_0 \times H_0 \times W_0}$ and one-hot-encoded labels $\mathbf{y}_i \in \mathbb{R}^{N_C}$ modeled as the realization of a joint distribution $\mathcal{P}_{(X,Y)}$ where $N_C$ is the number of classes in the dataset. The input data $\mathbf{x}_i$ is processed via a neural network $f_{\boldsymbol{\theta}}$ which is a parametric probabilistic model such that $\hat{\mathbf{y}}_i = f_{\boldsymbol{\theta}}(\mathbf{x}_i) = P(Y = \mathbf{y}_i | X = \mathbf{x}_i; \boldsymbol{\theta})$. This approach consists in considering the prediction $\hat{\mathbf{y}}_i$ as parameters of a Multinoulli distribution.

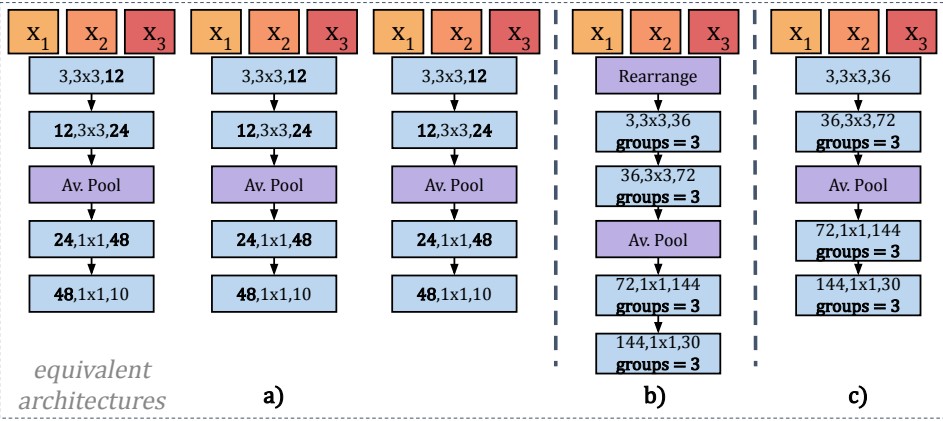

Figure 3: **Equivalent architectures for Packed-Ensembles.** **(a)** corresponds to the first sequential version, **(b)** to the version with the rearrange operation and grouped convolutions and **(c)** to the final version beginning with a full convolution.

To improve the quality of both predictions and estimated uncertainties, as well as the detection of OOD samples, Lakshminarayanan et al. (2017) propose to ensemble $M$ randomly initialized DNNs as a large predictor called Deep Ensembles. These ensembles can be seen as a discrete approximation of the intractable Bayesian marginalization on the weights, according to Wilson & Izmailov (2020). If we note $\{\boldsymbol{\theta}_m\}_{m=0}^{M-1}$ the set of trained weights for the $M$ DNNs, Deep Ensembles consists in averaging the predictions of these $M$ DNNs as in equation (3).

$$P(\mathbf{y}_i|\mathbf{x}_i, \mathcal{D}) = \frac{1}{M} \sum_{m=0}^{M-1} P(\mathbf{y}_i|\mathbf{x}_i, \boldsymbol{\theta}_m) \tag{3}$$

## 3 PACKED-ENSEMBLES

This section describes how to train multiple subnetworks using grouped convolution efficiently. Then, we explain how our new architectures are equivalent to training several networks in parallel.

### 3.1 REVISITING DEEP ENSEMBLES

Although Deep Ensembles provide undisputed benefits, they also come with the significant drawback that the training time and the memory usage in inference increase linearly with the number of networks. To alleviate these problems, we propose assembling small subnetworks, which are essentially DNNs with fewer parameters. Moreover, while ensembles to this day have mostly been trained sequentially, we suggest leveraging grouped convolutions to massively accelerate their training and inference computations thanks to their smaller size. The propagation of grouped convolutions with $M$ groups, $M$ being the number of subnetworks in the ensemble, ensures that the subnetworks are trained independently while dividing their encoding dimension by a factor $M$. More details on the usefulness of grouped convolutions to train ensembles can be found in subsection 3.3.

To create Packed-Ensembles (illustrated in Figure 2), we build on small subnetworks but compensate for the dramatic decrease of the model capacity by multiplying the width by the hyperparameter $\alpha$, which can be seen as an expansion factor. Hence, we propose *Packed-Ensembles-*$(\alpha, M, 1)$ as a flexible formalization of ensembles of small subnetworks. For an ensemble of $M$ subnetworks, Packed-Ensembles-$(\alpha, M, 1)$ therefore modifies the encoding dimension by a factor $\frac{\alpha}{M}$ and the inference of our ensemble is computed with the following formula, omitting the index $i$ of the sample:

$$\hat{\mathbf{y}} = \frac{1}{M} \sum_{m=0}^{M-1} P(\mathbf{y}|\boldsymbol{\theta}_{\alpha,m}, \mathbf{x}) \text{ with } \boldsymbol{\theta}_{\alpha,m} = \{\boldsymbol{\omega}_\alpha^j \circ \text{mask}_m^j\}_j, \tag{4}$$

where $\boldsymbol{\omega}^{j,\alpha}$ is the weight of the layer $j$ of dimension $(\alpha C_{j+1}) \times (\alpha C_j) \times s_j^2$.

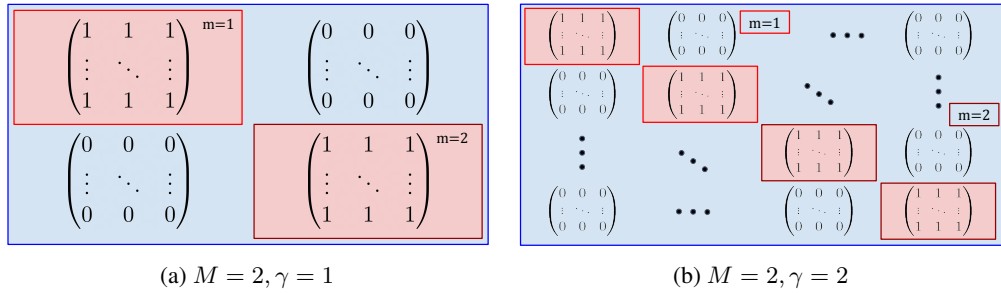

(a) $M = 2, \gamma = 1$             (b) $M = 2, \gamma = 2$

Figure 4: Diagram representation of a subnetwork mask: $\mathtt{mask}^j$, with $M = 2$, $j$ an integer corresponding to a fully connected layer

In the following, we introduce another hyperparameter $\gamma$ corresponding to the number of groups of each subnetwork of the Packed-Ensembles, creating another level of sparsity. These groups are also called "subgroups" and are applied to the different subnetworks. Formally, we denote our technique *Packed-Ensembles*-$(\alpha, M, \gamma)$, with the hyperparameters in the parentheses. In this work, we consider a constant number of subgroups across the layers; therefore, $\gamma$ divides $\alpha C_j$ for all $j$.

## 3.2 COMPUTATIONAL COST

For a convolutional layer, the number of parameters involving $C_j$ input channels, $C_{j+1}$ output channels, kernels of size $s_j$ and $\gamma$ subgroups is equal to $M \times \left[ \frac{\alpha C_j}{M} \frac{\alpha C_{j+1}}{M} s_j^2 \gamma^{-1} \right]$.

The same formula applies to dense layers as $1 \times 1$ convolutions. Two cases emerge when the architectures of the subnetworks are fully convolutional or dense. If $\alpha = \sqrt{M}$, the number of parameters in the ensemble equals the number of parameters in a single model. With $\alpha = M$, each subnetwork corresponds to a single model (and their ensemble is therefore equivalent in size to DE).

## 3.3 IMPLEMENTATION DETAILS

We propose a simple way of designing efficient ensemble convolutional layers using grouped convolutions. To take advantage of the parallelization capabilities of GPUs in training and inference, we replace the sequential training architecture, **(a)** in Figure 3, with the parallel implementations **(b)** and **(c)**. Figure 3 summarizes different equivalent architectures for a simple ensemble of $M = 3$ DNNs with three convolutional layers and a final dense layer (equivalent to a $1 \times 1$ convolution) with $\alpha = \gamma = 1$.

In **(b)**, we propose to stack the feature maps on the channel dimension (denoted as the $\mathtt{rearrange}$ operation).[1] This yields a feature map $\mathbf{h}^j$, of size $M \times C_j \times H_j \times W_j$ regrouped by batches of size only $\frac{B}{M}$, with $B$ the batch size of the ensemble. One solution to keep the same batch size is to repeat the batch $M$ times so that its size equals $B$ after the rearrangement. Using convolutions with $M$ groups and $\gamma$ subgroups per subnetwork, each feature map is convoluted separately by each subnetwork and yields its own independent output. Grouped convolutions are propagated until the end to ensure that gradients stay independent between subnetworks. Other operations, such as Batch Normalization (Ioffe & Szegedy, 2015), can be applied directly as long as they can be grouped or have independent actions on each channel. Figure 4a illustrates the mask used to code Packed-Ensembles in the case where $M = 2$. Similarly, Figure 4b shows the mask with $M = 2$ and $\gamma = 2$.

Finally, **(b)** and **(c)** are also equivalent. It is indeed possible to replace the $\mathtt{rearrange}$ operation and the first grouped convolution with a standard convolution if the same images are to be provided simultaneously to all the subnetworks. We confirm in Appendix F that this procedure is not detrimental to the ensemble's performance, and we take advantage of this property to provide this final optimization and simplification.

---

[1]See https://einops.rocks/api/rearrange/

## 4 EXPERIMENTS

To validate the performance of our method, we conduct experiments on classification tasks and measure the influence of the parameters $\alpha$ and $\gamma$. Regression tasks are detailed in Appendix N.

### 4.1 DATASETS AND ARCHITECTURES

First, we demonstrate the efficiency of Packed-Ensembles on CIFAR-10 and CIFAR-100 (Krizhevsky, 2009), showing how the method adapts to tasks of different complexities. As we propose to replace a single model architecture with several subnetworks, we study the behavior of PE on various sizes architectures: ResNet-18, ResNet-50 (He et al., 2016), and Wide ResNet28-10 (Zagoruyko & Komodakis, 2016). We compare it against Deep Ensembles (Lakshminarayanan et al., 2017) and three other approximated ensembles from the literature: BatchEnsemble (Wen et al., 2019), MIMO (Havasi et al., 2020), and Masksembles (Durasov et al., 2021).

Second, we report our results for Packed-Ensembles on ImageNet (Deng et al., 2009), which we compare against all baselines. We run experiments with ResNet-50 and ResNet-50x4. All training runs are started from scratch.

#### 4.1.1 METRICS, OOD DATASETS, AND IMPLEMENTATION

We evaluate the overall performance of the models in classification tasks using the accuracy (Acc) in % and the Negative Log-Likelihood (NLL). We choose the classical Expected Calibration Error (ECE) (Naeini et al., 2015) for the calibration of uncertainties[2] and measure the quality of the OOD detection using the Areas Under the Precision/Recall curve (AUPR) and Under the operating Curve (AUC), as well as the False Positive Rate at $95\%$ recall (FPR95), all expressed in %, similarly to Hendrycks & Gimpel (2017).

We use accuracy as the validation criterion (i.e., the final trained model is the one with the highest accuracy). During inference, we average the softmax probabilities of all subnetworks and consider the index of the maximum of the output vector to be the predicted class of the ensemble. We define the prediction confidence as this maximum value (also called maximum softmax probability).

For OOD detection tasks on CIFAR-10 and CIFAR-100, we use the SVHN dataset (Netzer et al., 2011) as an out-of-distribution dataset and transform the initial classification problem into a binary classification between in-distribution and OOD data using the maximum softmax probability as criterion. We discuss the different OOD criteria in appendix E. For ImageNet, we use two out-of-distribution datasets: ImageNet-O (Hendrycks et al., 2021b) and Texture (Wang et al., 2022), and use the Mutual Information (MI) as a criterion for the ensembles techniques (see Appendix E for details on MI) and the maximum softmax probability for the single model and MIMO. To measure the robustness under distribution shift, we use ImageNet-R (Hendrycks et al., 2021a) and evaluate the Accuracy, ECE, and NLL, denoted rAcc, rECE, and rNLL on this dataset, respectively.

We implement our models using the PyTorch-Lightning framework built on top of PyTorch. Both are open-source Python frameworks. Appendix B and Table 4 detail the hyper-parameters used in our experiments across architectures and datasets. Most training instances are completed on a single Nvidia RTX 3090 except for ImageNet, for which we use 2 to 8 Nvidia A100-80GB.

#### 4.1.2 RESULTS

Table 1 presents the average performance for the classification task over five runs using the hyper-parameters in Table 4. We demonstrate that Packed-Ensembles, in the setting of $\alpha = 2$ and $\gamma = 2$, yields similar results to Deep Ensembles while having a lower memory cost than a single model. For CIFAR-10, the relative performance of PE compared to DE appears to increase as the original architecture becomes larger. When using ResNet-18, Packed-Ensembles matches Deep Ensembles on OOD detection metrics but shows slightly worse performance on the others. However, using ResNet-50, both models seem to perform similarly, and PE slightly outperforms DE in classification performance with WideResNet28-10.

---

[2]Note that the benchmark uncertainty-baselines only uses ECE to measure calibration

Table 1: **Performance comparison (averaged over five runs) on CIFAR-10/100 using ResNet-18 (R18), ResNet-50 (R50), and Wide ResNet28-10 (WR) architectures.** All ensembles have $M = 4$ subnetworks, we highlight the best performances in bold. For our method, we consider $\alpha = \gamma = 2$, except for WR on C100, where $\gamma = 1$. *Mult-Adds* corresponds to the inference cost, i.e., the number of Giga multiply-add operations for a forward pass which is estimated with Torchinfo (2022).

| Method | Data | Net | Acc ↑ | NLL ↓ | ECE ↓ | AUPR ↑ | AUC ↑ | FPR95 ↓ | Params (M) ↓ | Mult-Adds ↓ |
|---|---|---|---|---|---|---|---|---|---|---|
| Single Model | C10 | R18 | 94.0 | 0.238 | 0.035 | 94.0 | 89.7 | 33.8 | 11.17 | 0.56 |
| BatchEnsemble | C10 | R18 | 92.9 | 0.257 | 0.031 | 92.4 | 87.8 | 32.1 | 11.21 | 2.22 |
| MIMO ($\rho = 1$) | C10 | R18 | 94.0 | 0.228 | 0.033 | 94.4 | 90.2 | 28.6 | 11.19 | 0.56 |
| Masksembles | C10 | R18 | 94.0 | 0.188 | 0.009 | 93.6 | 89.5 | 27.8 | 11.24 | 2.22 |
| Packed-Ensembles | C10 | R18 | 94.3 | 0.178 | **0.007** | **94.7** | **91.3** | 23.2 | **8.18** | **0.48** |
| Deep Ensembles | C10 | R18 | **95.1** | **0.156** | 0.008 | **94.7** | **91.3** | **18.0** | 44.70 | 2.22 |
| Single Model | C10 | R50 | 95.1 | 0.211 | 0.031 | 95.2 | 91.9 | 23.6 | 23.52 | 1.30 |
| BatchEnsemble | C10 | R50 | 93.9 | 0.255 | 0.033 | 94.7 | 91.3 | 20.1 | 23.63 | 5.19 |
| MIMO ($\rho = 1$) | C10 | R50 | 95.4 | 0.197 | 0.030 | 95.1 | 90.8 | 26.0 | 23.59 | 1.30 |
| Masksembles | C10 | R50 | 95.3 | 0.175 | 0.019 | 95.7 | 92.2 | 22.1 | 23.81 | 5.19 |
| Packed-Ensembles | C10 | R50 | 95.9 | 0.137 | **0.008** | **97.3** | **95.2** | **14.4** | **14.55** | **1.00** |
| Deep Ensembles | C10 | R50 | **96.0** | **0.136** | **0.008** | 97.0 | 94.7 | 15.5 | 94.08 | 5.19 |
| Single Model | C10 | WR | 95.4 | 0.200 | 0.029 | 96.1 | 93.2 | 20.4 | 36.49 | 5.95 |
| BatchEnsemble | C10 | WR | 95.6 | 0.206 | 0.027 | 95.5 | 92.5 | 22.1 | 36.59 | 23.81 |
| MIMO ($\rho = 1$) | C10 | WR | 94.7 | 0.234 | 0.034 | 94.9 | 90.6 | 30.9 | 36.51 | 5.96 |
| Masksembles | C10 | WR | 94.0 | 0.186 | 0.016 | 97.2 | 95.0 | 14.5 | 36.53 | 23.82 |
| Packed-Ensembles | C10 | WR | **96.2** | **0.133** | 0.009 | **98.1** | **96.5** | **11.1** | 19.35 | 4.06 |
| Deep Ensembles | C10 | WR | 95.8 | 0.143 | 0.013 | 97.8 | 96.0 | 12.5 | 145.96 | 23.82 |
| Single Model | C100 | R18 | 75.1 | 1.016 | 0.093 | 88.6 | 79.5 | 55.0 | 11.22 | 0.56 |
| BatchEnsemble | C100 | R18 | 71.2 | 1.236 | 0.116 | 86.0 | 75.4 | 60.2 | 11.25 | 2.22 |
| MIMO ($\rho = 1$) | C100 | R18 | 75.3 | 0.962 | 0.069 | 89.2 | 80.7 | 52.9 | 11.36 | 0.56 |
| Masksembles | C100 | R18 | 74.2 | 1.054 | 0.061 | 86.7 | 76.3 | 59.8 | 11.24 | 2.22 |
| Packed-Ensembles | C100 | R18 | 76.4 | 0.858 | 0.041 | 88.7 | 79.8 | 57.1 | **8.27** | **0.48** |
| Deep Ensembles | C100 | R18 | **78.2** | **0.800** | **0.018** | **90.2** | **82.4** | **50.5** | 44.88 | 2.22 |
| Single Model | C100 | R50 | 78.3 | 0.905 | 0.089 | 87.4 | 77.9 | 57.6 | 23.70 | 1.30 |
| BatchEnsemble | C100 | R50 | 66.6 | 1.788 | 0.182 | 85.2 | 74.6 | 60.6 | 23.81 | 5.19 |
| MIMO ($\rho = 1$) | C100 | R50 | 79.0 | 0.876 | 0.079 | 87.5 | 76.9 | 64.7 | 24.33 | 1.30 |
| Masksembles | C100 | R50 | 78.5 | 0.832 | 0.046 | 90.3 | 81.9 | 52.3 | 23.81 | 5.19 |
| Packed-Ensembles | C100 | R50 | **81.2** | **0.703** | **0.020** | **90.0** | **81.7** | 56.5 | **15.55** | **1.00** |
| Deep Ensembles | C100 | R50 | 80.9 | 0.713 | 0.026 | 89.2 | 80.8 | **52.5** | 94.82 | 5.19 |
| Single Model | C100 | WR | 80.3 | 0.963 | 0.156 | 81.0 | 64.2 | 80.1 | **36.55** | **5.95** |
| BatchEnsemble | C100 | WR | 82.3 | 0.835 | 0.130 | **88.1** | **78.2** | **69.8** | 36.65 | 23.81 |
| MIMO ($\rho = 1$) | C100 | WR | 80.2 | 0.822 | **0.028** | 84.9 | 72.0 | 72.8 | 36.74 | 5.96 |
| Masksembles | C100 | WR | 74.4 | 0.937 | 0.063 | 76.1 | 60.0 | 75.1 | 36.59 | 23.82 |
| Packed-Ensembles | C100 | WR | **83.9** | **0.678** | 0.089 | 86.2 | 73.2 | 80.7 | 36.62 | **5.95** |
| Deep Ensembles | C100 | WR | 82.5 | 0.903 | 0.229 | 81.6 | 67.9 | 71.3 | 146.19 | 23.82 |

On CIFAR-100, Deep Ensembles outperforms Packed-Ensembles on ResNet-18. However, we argue that ResNet-18 architecture need more representation capacity to be divided into subnetworks for CIFAR-100. Indeed, when we look at the results of ResNet-50, we can see that Packed-Ensembles has better results than Deep Ensembles. This analysis demonstrates that, given a sufficiently large network, Packed-Ensembles is able to match Deep Ensembles with only $16\%$ of its parameters. In Appendix D, we discuss the influence of the representation capacity.

Based on the results in Table 2, we can conclude that Packed-Ensembles improves uncertainty quantification for OOD and distribution shift on ImageNet compared to Deep Ensembles and Single model and that it improves the accuracy with a moderate training and inference cost.

### 4.1.3 STUDY ON THE PARAMETERS $\alpha$ AND $\gamma$

Table 1 reports results for $\alpha = 2$ and $\gamma = 2$. However, the optimal values of these hyperparameters depend on the balance between computational cost and performance. To help users strike the best compromise, we propose Figures 6 and 7 in Appendix D, which illustrate the impact of changing $\alpha$ on the performance of Packed-Ensembles.

## 5 DISCUSSIONS

We have shown that Packed-Ensembles has attractive properties, mainly by providing a similar quality of uncertainty quantification as Deep Ensembles while using a reduced architecture and

Table 2: **Performance comparison on ImageNet using ResNet-50 (R50) and ResNet-50x4 (R50x4).** All ensembles have $M = 4$ subnetworks and $\gamma = 1$. We highlight the best performances in bold. For OOD tasks, we use ImageNet-O (IO) and Texture (T), and for distribution shift we use ImageNet-R. The number of parameters and operations are available in Appendix M.

| Method | Net | Acc | ECE | AUPR - T | AUC - T | FPR95 - T | AUPR - IO | AUC - IO | FPR95 - IO | rAcc | rNLL | rECE |
|---|---|---|---|---|---|---|---|---|---|---|---|---|
| Single Model | R50 | 77.8 | 0.121 | 18.0 | 80.9 | 68.6 | 3.6 | 50.8 | 90.8 | 23.5 | 5.187 | 0.082 |
| BatchEnsemble | R50 | 75.9 | 0.035 | 20.2 | 81.6 | 66.5 | 4.0 | 55.2 | 82.3 | 21.0 | 6.148 | 0.165 |
| MIMO ($\rho = 1$) | R50 | 77.6 | 0.147 | 18.4 | 81.6 | 66.8 | 3.7 | 52.2 | 90.6 | 23.4 | 5.115 | 0.059 |
| Masksembles | R50 | 73.6 | 0.209 | 13.6 | 79.7 | 68.3 | 3.3 | 47.7 | 87.7 | 21.2 | 5.139 | 0.011 |
| Packed-Ensembles $\alpha = 3$ | R50 | 77.9 | 0.180 | **35.1** | **88.2** | **43.7** | **9.9** | **68.4** | 80.9 | 23.8 | 4.978 | 0.022 |
| Deep Ensembles | R50 | **79.2** | 0.233 | 19.6 | 83.4 | 62.1 | 3.7 | 52.5 | 85.5 | **24.9** | **4.879** | **0.018** |
| Single Model | R50×4 | 80.2 | 0.022 | 20.5 | 82.6 | 63.9 | 4.9 | 60.2 | 87.4 | 26.0 | 5.190 | 0.1721 |
| BatchEnsemble | R50×4 | 77.7 | 0.024 | 23.8 | 82.8 | 63.8 | 4.4 | 58.4 | 80.5 | 23.4 | 6.079 | 0.203 |
| MIMO ($\rho = 1$) | R50×4 | 80.3 | 0.015 | 19.3 | 82.5 | 66.1 | 4.9 | 60.7 | 86.4 | 25.8 | 5.278 | 0.189 |
| Masksembles | R50×4 | 79.8 | 0.137 | 21.5 | 83.3 | 63.5 | 4.4 | 58.4 | 80.5 | 23.4 | 6.079 | 0.207 |
| Packed-Ensembles $\alpha = 2$ | R50×4 | 81.3 | 0.103 | **34.6** | **88.1** | **50.3** | **9.6** | **69.9** | 79.2 | 26.6 | 4.848 | **0.075** |
| Deep Ensembles | R50×4 | **82.1** | 0.053 | 23.0 | 85.6 | 58.1 | 5.0 | 62.7 | 81.9 | **28.2** | **4.789** | 0.105 |

computing cost. Several questions can be raised, and we conducted some studies - detailed in the Appendix sections - to provide possible answers.

**Discussion on the sparsity** As described in section 3, one could interpret PE as leveraging group convolutions to approximate Deep Ensembles with a mask operation applied to some components. In Appendix C, by using a simplified model, we propose a bound of the approximation error based on the Kullback-Leibler divergence between the DE and its pruned version. This bound depends on the density of ones in the mask $p$, and, more specifically, depends on $p(1 - p)$ and $(1 - p)^2/p$. By manipulating these terms, corresponding to modifying the number of subnetworks $M$, the number of groups $\gamma$, and the dilation factor $\alpha$, we could theoretically control the approximation error.

**On the sources of stochasticity** Diversity is essential in ensembles and is usually obtained by exploiting two primary sources of stochasticity: the random initialization of the model's parameters and the shuffling of the batches. A last source of stochasticity is introduced during training by the non-deterministic behavior of the backpropagation algorithms. In Appendix F, we study the function space diversities which arise from every possible combination of these sources. It follows that only one of these sources is often sufficient to generate diversity, and no peculiar pattern seems to emerge to predict the best combination. Specifically, we highlight that even the only use of non-deterministic algorithms introduces enough diversity between each subnetwork of the ensemble.

**Ablation study** We perform ablation studies to assess the impact of the parameters $M$, $\alpha$, and $\gamma$ on the performance of Packed-Ensembles. Appendix D provides in-depth details of this study. No explicit behavior appears from the results we obtained. A trend shows that a higher number of subnetworks helps get better OOD detection, but the improvement in AUPR is not significant.

**Training speed** Depending on the chosen hyperparameters $\alpha$, $M$, and $\gamma$, PE may have fewer parameters than the single model, as shown in Table 1. This translates into an expected lower number of operations. A study of the training and inference speeds, developed in Appendix H, shows that using PE-(2,4,1) does not significantly increase the training and testing times compared to the single model while improving accuracy and uncertainty quantification performances. However, this also hints that the group-convolution speedup is not optimal despite the significant acceleration offered by 16-bits floating points.

**OOD criteria** The maximum softmax probability is often used as criterion for discriminating OOD elements. However, this criterion is not unique, and others can be used, such as the Mutual Information, the maximum logit, or the Shannon entropy of the mean prediction. Although no relationship is expected between this criterion and PE, we obtained different performances in OOD detection according to the selected criterion. The results on CIFAR-100 are detailed in Appendix E and show that an approach based on the maximum logit seems to give the best results in detecting OOD. It should be noted that the notion of OOD depends on the training distribution. Such a discussion does not necessarily generalize to all datasets. Indeed, preliminary results have shown that Mutual information outperforms the other criteria for our method applied to the ImageNet dataset.

## 6    RELATED WORK

**Ensembles and uncertainty quantification.**    Bayesian Neural Networks (BNNs) (MacKay, 1992; Neal, 1995) are the cornerstone and primary source of inspiration for uncertainty quantification in deep learning. Despite the progress enabled by variational inference (Jordan et al., 1999; Blundell et al., 2015), BNNs remain challenging to scale and train for large DNN architectures (Dusenberry et al., 2020). DE (Lakshminarayanan et al., 2017) arise as a practical and efficient instance of BNNs, coarsely but effectively approximating the posterior distribution of weights (Wilson & Izmailov, 2020). DE are currently the best-performing approach for both predictive performance and uncertainty estimation (Ovadia et al., 2019; Gustafsson et al., 2020).

**Efficient ensembles.**    The appealing properties in performance and diversity of DE (Fort et al., 2019), but also their major downside related to computational cost, have inspired a large cohort of approaches aiming to mitigate it. BatchEnsemble (Wen et al., 2019) spawns an ensemble at each layer thanks to an efficient parameterization of subnetwork-specific parameters trained in parallel. MIMO (Havasi et al., 2020) shows that a large network can encapsulate multiple subnetworks using a multi-input multi-output configuration. A single network can be used in ensemble mode by disabling different sub-sets of weights at each forward pass  (Gal & Ghahramani, 2016; Durasov et al., 2021). Liu et al. (2022) leverage the sparse networks training algorithm of Mocanu et al. (2018) to produce ensembles of sparse networks. Ensembles can be computed from a single training run by collecting intermediate model checkpoints (Huang et al., 2017; Garipov et al., 2018), by computing the posterior distribution of the weights by tracking their trajectory during training (Maddox et al., 2019; Franchi et al., 2020), and by ensembling predictions over multiple augmentations of the input sample (Ashukha et al., 2020). However, most of these approaches require multiple forward passes.

**Neural network compression.**    The most intuitive approach for reducing the size of a model is to employ DNNs that are memory-efficient by design, relying on, e.g., channel shuffling (Zhang & Yang, 2021), point-wise convolutional filters (Liang et al., 2021), weight sharing (Bender et al., 2020), or a combination of them. Some of the most popular architectures that leverage such models are SqueezeNet (Iandola et al., 2016), ShuffleNet (Zhang et al., 2018b), and MobileNet-v3 (Howard et al., 2019). Some approaches conduct automatic model size reduction, e.g., network sparsification (Molchanov et al., 2017; Louizos et al., 2018; Frankle & Carbin, 2018; Tartaglione et al., 2022). These approaches aim at removing as many parameters as possible from the model to improve memory and computation efficiency, also at train time Bragagnolo et al. (2022). Similarly, quantization approaches (Han et al., 2016; Lin et al., 2017) avoid or minimize the computation cost of floating point operation and optimize the use of the much more efficient integer computation.

**Grouped convolutions.**    To the best of our knowledge, grouped convolutions (group of convolutions) were introduced by Krizhevsky et al. (2012). Enabling the computation of several independent convolutions in parallel, they developed the idea of running a single model on multiple GPU devices. Xie et al. (2017) demonstrate that using grouped convolutions leads to accuracy improvements and model complexity reduction. So far, grouped convolutions have been used primarily for computational efficiency but also to compute multiple output branches in parallel (Chen & Shrivastava, 2020). PE re-purpose them to delineate multiple subnetworks within a network and efficiently train an ensemble of such subnetworks.

## 7    CONCLUSIONS

We propose a new ensemble framework, Packed-Ensembles, that can approximate Deep Ensembles in terms of uncertainty quantification and accuracy. Our research provides several new findings. First, we show that small independent neural networks can be as effective as large, deep neural networks when used in ensembles. Secondly, we demonstrate that not all sources of diversity are essential for improving ensemble diversity. Thirdly, we show that Packed-Ensembles are more stable than single DNNs. Fourthly, we highlight that there is a trade-off between accuracy and the number of parameters, and Packed-Ensembles enables us to create flexible and efficient ensembles.

In the future, we intend to explore Packed-Ensembles for more complex downstream tasks.

## 8 REPRODUCIBILITY

Alongside this paper, we provide the source code of Packed-Ensembles layers. Additionally, we have created two notebooks demonstrating how to train ResNet-50-based Packed-Ensembles using public datasets such as CIFAR-10 and CIFAR-100. To ensure reproducibility, we report the performance given a specific random seed with a deterministic training process. Furthermore, it should be noted that the source code contains two PyTorch Module classes to produce Packed-Ensembles efficiently. A readme file at the root of the project details how to install and run experiments. In addition, we showcase how to get Packed-Ensembles from LeNet (LeCun et al., 1998).

To further promote accessibility, we have created an open-source pip-installable PyTorch package, torch-uncertainty, that includes Packed-Ensembles layers. With these resources, we hope to encourage the broader research community to engage with and build upon our work.

## 9 ETHICS

The purpose of this paper is to propose a new method for better estimations of uncertainty for deep-learning-based models. Nevertheless, we acknowledge their limitations, which could become particularly concerning when applied to safety-critical systems. While this work aims to improve the reliability of Deep Neural Networks, this approach is not ready for deployment in safety-critical systems. We show the limitations of our approach in several experiments. Many more validation and verification steps would be crucial before considering its real-world implementation to ensure robustness to various unknown situations, including corner cases, adversarial attacks, and potential biases.

### ACKNOWLEDGMENTS

This work was supported by AID Project ACoCaTherm and Hi!Paris. This work was performed using HPC resources from GENCI-IDRIS (Grant 2021-AD011011970R1) and (Grant 2022-AD011011970R2).

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

TABLE OF CONTENTS - SUPPLEMENTARY MATERIAL

# A  NOTATIONS

We summarize the main notations used in the paper in Table 3.

Table 3: **Summary of the main notations of the paper.**

| Notations | Meaning |
|---|---|
| $\mathcal{D} = \{(\mathbf{x}_i, \mathbf{y}_i)\}_{i=1}^{|\mathcal{D}|}$ | The set of $|\mathcal{D}|$ data samples and the corresponding labels |
| $j, m, L$ | The index of the current layer, the current subnetwork, and the number of layers |
| $\mathbf{z}^j$ | The preactivation feature map and output of the layer $(j-1)$/input of layer $j$ |
| $\phi$ | The activation function (considered constant throughout the network) |
| $\mathbf{h}^j$ | The feature map and output of layer $j$, $\mathbf{h}^j = \phi(\mathbf{z}^j)$ |
| $H_j, W_j$ | The height and width of the feature maps and output of layer $j-1$ |
| $C_j$ | The number of channels of the feature maps and output of layer $j-1$ |
| $n_j$ | The number of parameters of layer $j$ |
| B | The batch size of the training procedure |
| $\text{mask}_m^j$ | The mask corresponding to the layer $j$ of the subnetwork $m$ |
| $\lfloor \cdot \rfloor$ | The floor function |
| $\star, \circledast, \circ$ | The 2D cross-correlation, the convolution, and the Hadamard product |
| $s_j$ | The size of the kernel of the layer $j$ |
| $M$ | The number of subnetworks in an ensemble |
| $\hat{\mathbf{y}}_i^m$ | The prediction of the subnetwork number $m$ concerning the input $\mathbf{x}_i$ |
| $\hat{\mathbf{y}}_i$ | The prediction of the ensemble concerning the input $\mathbf{x}_i$ |
| $\alpha$ | The width-augmentation factor of Packed-Ensembles |
| $\gamma$ | The number of subgroups of Packed-Ensembles |
| $\boldsymbol{\theta}_{\alpha,m}$ | The set of weights of the subnetwork $m$ with a width factor $\alpha$ |
| $\boldsymbol{\omega}_{\alpha,\gamma}^j$ | The weights of layer $j$ with $\gamma$ groups and a width factor $\alpha$ |

Table 4: **Hyperparameters for image classification experiments.** HFlip denotes the classical horizontal flip.

| Dataset | Networks | Epochs | Batch size | start lr | Momentum | Weight decay | $\gamma$-lr | Milestones | Data augmentations |
|---------|----------|--------|-----------|----------|----------|--------------|-------------|------------|--------------------|
| C10 | R18 | 75 | 128 | 0.05 | 0.9 | 5e-4 | 0.1 | 25, 50 | HFlip |
| C10 | R50 | 200 | 128 | 0.1 | 0.9 | 5e-4 | 0.2 | 60, 120, 160 | HFlip |
| C10 | WR28-10 | 200 | 128 | 0.1 | 0.9 | 5e-4 | 0.2 | 60, 120, 160 | HFlip |
| C100 | R18 | 75 | 128 | 0.05 | 0.9 | 1e-4 | 0.2 | 25, 50 | HFlip |
| C100 | R50 | 200 | 128 | 0.1 | 0.9 | 5e-4 | 0.2 | 60, 120, 160 | HFlip |
| C100 | WR28-10 | 200 | 128 | 0.1 | 0.9 | 5e-4 | 0.2 | 60, 120, 160 | Medium |

# B  IMPLEMENTATION DETAILS

**General Considerations.**  Table 4 summarizes all the hyperparameters used in the paper for CIFAR-10 and CIFAR-100. In all cases, we use SGD combined with a multistep-learning-rate scheduler multiplying the rate by $\gamma$-lr at each milestone. Note that BatchEnsemble based on ResNet-50 uses a lower learning rate of 0.08 instead of 0.1 for stability. The "Medium" data augmentation corresponds to a combination of mixup (Zhang et al., 2018a) and cutmix (Yun et al., 2019) with 0.5 switch probability and using timm's augmentation classes (Wightman, 2019), with coefficients respectively 0.5 and 0.2. In this case, we also use RandAugment (Cubuk et al., 2020) with $m = 9$, $n = 2$, and $mstd = 1$ and a label-smoothing (Szegedy et al., 2016) of intensity 0.1.

To ensure that the layers convey sufficient information and are not weakened by groups, we have set a constant minimum number of channels per group to 64 for all experiments presented in the paper. If the number of channels per group is lower than this threshold, $\gamma$ is reduced. Moreover, we do not apply subgroups (parameterized by $\gamma$) on the first layer of the network, nor on the first layer of ResNet's blocks. Experiments in which this minimum number of channels could play a significant role and bring confusion are not presented (see, for instance, PE-$(1, 4, 4)$ in Table 5).

For ImageNet, we use the A3 procedure from Wightman et al. (2021) for all models. Training with the exact A3 procedure was not always possible. Refer to the specific subsection for more details.

Please note that the hyperparameters of the training procedures have not been optimized for our method and have been taken directly from the literature (He et al., 2016; Wightman et al., 2021). We strengthened the data augmentations for WideResNet on CIFAR-100 as we were not able to replicate the results from Zagoruyko & Komodakis (2016).

**Masksembles.**  We use the code proposed by (Durasov et al., 2021) [3]. We modified the mask generation function using binary search, as proposed by the authors since it was unable to build masks for ResNet50x4. We note that the code implies performing batch repeats at the start of the forward passes. All the results regarding this technique are therefore computed with this specification. The ResNet implementations are built using Masksemble2D layers with $M = 4$ and a scale factor of 2 after each convolution.

**BatchEnsemble.**  For BatchEnsemble, we use two different values for weight decay: table 4 provides the weight decay corresponding to the shared weights but we don't apply weight decay to the vectors S and R (which generate the rank-1 matrices).

**ImageNet.**  The batch size of Masksembles ResNet-50x4 is reduced to 1120 because of memory constraints. Concerning the BatchEnsembles based on ResNet-50 and ResNet-50x4, we clip the norm of the gradients to 0.0005 to avoid divergence.

# C  DISCUSSION ON THE SPARSITY

In this section, we estimate the expected distance between a dense, fully-connected layer and a sparse one. For simplicity, we are here assuming to operate with a fully-connected layer. First, let us write our first proposition:

**Proposition C.1.** *Given a fully connected layer $j + 1$ defined by:*

$$\mathbf{z}^{j+1}(c) = \sum_{k=0}^{C_j - 1} \boldsymbol{\omega}^j(c, k) \mathbf{h}^j(k) \tag{5}$$

---

[3]available at github.com/nikitadurasov/masksembles

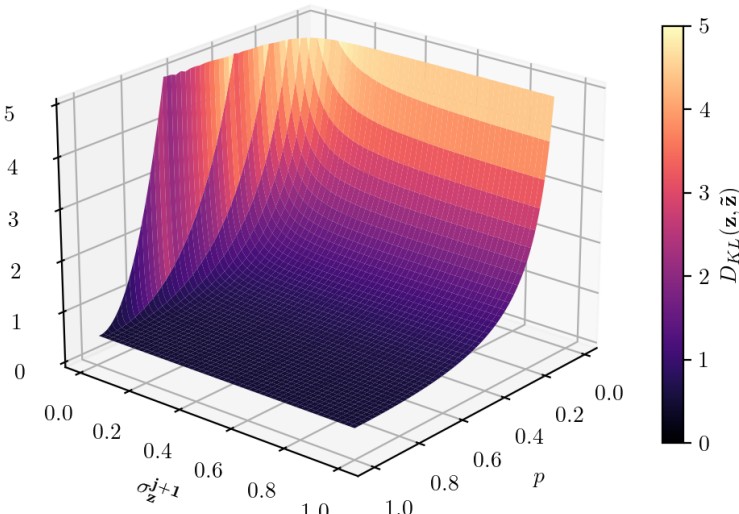

Figure 5: KL divergence for different values of $p$ and $\boldsymbol{\sigma_z^{j+1}}$, with $\boldsymbol{\mu^j}(k) = 0.1 \ \forall j, k$ and $w^j(c, k) = 0.1 \ \forall j, c, k$.

*and its approximation defined by:*

$$\tilde{\mathbf{z}}^{j+1}(c) = \sum_{k=0}^{C_j-1} (\boldsymbol{\omega}^j(c, k) mask^j(k, c)) \mathbf{h}^j(k) \tag{6}$$

*Under the assumption that the $j$ follows a Gaussian distribution $\boldsymbol{h}^j \sim \mathcal{N}(\boldsymbol{\mu^j}, \Sigma^j)$, where $\Sigma^j$ is the covariance matrix, and $\boldsymbol{\mu^j}$ the mean vector, the Kullback–Leibler divergence between the layer and its approximation is bounded by:*

$$D_{\mathrm{KL}}(\mathbf{z}, \tilde{\mathbf{z}})(c) \leq \frac{1}{2} \left\{ p + \frac{1}{p} - 2 + \frac{p \cdot (1-p) \sum_{k=0}^{C_j-1} \boldsymbol{\omega}^j(c, k)^2 \boldsymbol{\mu^j}(k)^2}{(\boldsymbol{\sigma_z^{j+1}})^2(c)} + \frac{\left[(1-p) \times \boldsymbol{\mu_z^{j+1}}(c)\right]^2}{p(\boldsymbol{\sigma_z^{j+1}})^2(c)} \right\} \tag{7}$$

*where $p \in [0; 1]$ is the fraction of the parameters of $\mathbf{z}^{j+1}(c)$ included in the approximation $\tilde{\mathbf{z}}^{j+1}(c)$.*

A plot for (7) is provided in Figure 5.

*Proof.* To prove Prop. C.1, we state first that, since $\mathbf{h}^j(k)$ follows a Gaussian distribution, and considering that $\boldsymbol{\omega}^j$ at inference time is constant and linearly-combined with a gaussian random variable, $z^{j+1}$ will be as well gaussian-distributed.
From the property of linearity of expectations, we know that the mean for $\mathbf{z}^{j+1}(c)$ is:

$$\boldsymbol{\mu_z^{j+1}}(c) = \sum_{k=0}^{C_j-1} \boldsymbol{\omega}^j(c, k) \boldsymbol{\mu^j}(k) \tag{8}$$

and the variance is:

$$(\boldsymbol{\sigma_z^{j+1}})^2(c) = \sum_{k=0}^{C_j-1} \boldsymbol{\omega}^j(c, k) \left[ \boldsymbol{\omega}^j(c, k) \Sigma(k, k) + 2 \sum_{k'<k} \boldsymbol{\omega}^j(c, k') \Sigma(k', k) \right]. \tag{9}$$

If we assume $\Sigma(i, k) = 0 \ \forall \ i \neq k$, (9) simplifies into:

$$(\boldsymbol{\sigma_z^{j+1}})^2(c) = \sum_{k=0}^{C_j-1} \boldsymbol{\omega}^j(c, k)^2 \Sigma(k, k). \tag{10}$$

Let us now consider the case with the mask, similarly as presented at the end of section 2.1:

$$\tilde{\mathbf{z}}^{j+1}(c) = \sum_{k=0}^{C_j-1} (\boldsymbol{\omega}^j(c,k)\text{mask}^j(k,c))\mathbf{h}^j(k) \tag{11}$$

We assume here that $\text{mask}^j \sim Ber(p)$ where $p$ is the probability of the Bernoulli (or 1-pruning rate). In the limit of large $C_j$, we know that $\tilde{\mathbf{z}}^{j+1}(c)$ follows a Gaussian distribution defined by a mean and a variance equal to:

$$\tilde{\boldsymbol{\mu}}_{\mathbf{z}}^{j+1}(c) = \sum_{k=0}^{C_j-1} \boldsymbol{\omega}^j(c,k)\boldsymbol{\mu}^j(k)p \tag{12}$$

$$(\tilde{\boldsymbol{\sigma}}_{\mathbf{z}}^{j+1})^2(c) = \sum_{k=0}^{C_j-1} p\boldsymbol{\omega}^j(c,k)^2\left[\boldsymbol{\mu}^j(k)^2(1-p) + \Sigma(k,k)\right] \tag{13}$$

Hence, we have:

$$\tilde{\boldsymbol{\mu}}_{\mathbf{z}}^{j+1}(c) = p \times \boldsymbol{\mu}_{\mathbf{z}}^{j+1}(c) \tag{14}$$

$$(\tilde{\boldsymbol{\sigma}}_{\mathbf{z}}^{j+1})^2(c) = p\left[(\boldsymbol{\sigma}_{\mathbf{z}}^{j+1})^2(c) + (1-p)\sum_{k=0}^{C_j-1} \boldsymbol{\omega}^j(c,k)^2\boldsymbol{\mu}^j(k)^2\right] \tag{15}$$

In order to assess the dissimilarity between $\mathbf{z}$ and $\tilde{\mathbf{z}}$, we can write the Kullback–Leibler divergence:

$$D_{\text{KL}}(\mathbf{z},\tilde{\mathbf{z}})(c) = \frac{1}{2}\left\{ \log\left[\frac{(\tilde{\boldsymbol{\sigma}}_{\mathbf{z}}^{j+1})^2(c)}{(\boldsymbol{\sigma}_{\mathbf{z}}^{j+1})^2(c)}\right] + \frac{(\boldsymbol{\sigma}_{\mathbf{z}}^{j+1})^2(c) + \left[\boldsymbol{\mu}_{\mathbf{z}}^{j+1}(c) - \tilde{\boldsymbol{\mu}}_{\mathbf{z}}^{j+1}(c)\right]^2}{(\tilde{\boldsymbol{\sigma}}_{\mathbf{z}}^{j+1})^2(c)} - 1\right\} \tag{16}$$

Straightforwardly we can write the inequality:

$$D_{\text{KL}}(\mathbf{z},\tilde{\mathbf{z}})(c) \leq \frac{1}{2}\left\{ \frac{(\tilde{\boldsymbol{\sigma}}_{\mathbf{z}}^{j+1})^2(c)}{(\boldsymbol{\sigma}_{\mathbf{z}}^{j+1})^2(c)} - 1 + \frac{(\boldsymbol{\sigma}_{\mathbf{z}}^{j+1})^2(c) + \left[\boldsymbol{\mu}_{\mathbf{z}}^{j+1}(c) - \tilde{\boldsymbol{\mu}}_{\mathbf{z}}^{j+1}(c)\right]^2}{(\tilde{\boldsymbol{\sigma}}_{\mathbf{z}}^{j+1})^2(c)} - 1\right\} \tag{17}$$

According to (15) we can write:

$$D_{\text{KL}}(\mathbf{z},\tilde{\mathbf{z}})(c) \leq \frac{1}{2}\left\{ \frac{p\left[(\boldsymbol{\sigma}_{\mathbf{z}}^{j+1})^2(c) + (1-p)\sum_{k=0}^{C_j-1}\boldsymbol{\omega}^j(c,k)^2\boldsymbol{\mu}^j(k)^2\right]}{(\boldsymbol{\sigma}_{\mathbf{z}}^{j+1})^2(c)} - 1 + \right.$$
$$\left. + \frac{(\boldsymbol{\sigma}_{\mathbf{z}}^{j+1})^2(c) + \left[\boldsymbol{\mu}_{\mathbf{z}}^{j+1}(c) - \tilde{\boldsymbol{\mu}}_{\mathbf{z}}^{j+1}(c)\right]^2}{p\left[(\boldsymbol{\sigma}_{\mathbf{z}}^{j+1})^2(c) + (1-p)\sum_{k=0}^{C_j-1}\boldsymbol{\omega}^j(c,k)^2\boldsymbol{\mu}^j(k)^2\right]} - 1\right\} \tag{18}$$

Since we know that $\frac{(\boldsymbol{\sigma}_{\mathbf{z}}^{j+1})^2(c)+\left[\boldsymbol{\mu}_{\mathbf{z}}^{j+1}(c)-\tilde{\boldsymbol{\mu}}_{\mathbf{z}}^{j+1}(c)\right]^2}{p\left[(\boldsymbol{\sigma}_{\mathbf{z}}^{j+1})^2(c)+(1-p)\sum_{k=0}^{C_j-1}\boldsymbol{\omega}^j(c,k)^2\boldsymbol{\mu}^j(k)^2\right]} \leq \frac{(\boldsymbol{\sigma}_{\mathbf{z}}^{j+1})^2(c)+\left[\boldsymbol{\mu}_{\mathbf{z}}^{j+1}(c)-\tilde{\boldsymbol{\mu}}_{\mathbf{z}}^{j+1}(c)\right]^2}{p(\boldsymbol{\sigma}_{\mathbf{z}}^{j+1})^2(c)}$ we can also write:

$$D_{\text{KL}}(\mathbf{z},\tilde{\mathbf{z}})(c) \leq \frac{1}{2}\left\{ p - 1 + \frac{p\cdot(1-p)\sum_{k=0}^{C_j-1}\boldsymbol{\omega}^j(c,k)^2\boldsymbol{\mu}^j(k)^2}{(\boldsymbol{\sigma}_{\mathbf{z}}^{j+1})^2(c)} + \right.$$
$$\left. + \frac{(\boldsymbol{\sigma}_{\mathbf{z}}^{j+1})^2(c) + \left[\boldsymbol{\mu}_{\mathbf{z}}^{j+1}(c) - \tilde{\boldsymbol{\mu}}_{\mathbf{z}}^{j+1}(c)\right]^2}{p(\boldsymbol{\sigma}_{\mathbf{z}}^{j+1})^2(c)} - 1\right\} \tag{19}$$

Finally, according to: (14)

$$D_{\text{KL}}(\mathbf{z},\tilde{\mathbf{z}})(c) \leq \frac{1}{2}\left\{ p + \frac{1}{p} - 2 + \frac{p\cdot(1-p)\sum_{k=0}^{C_j-1}\boldsymbol{\omega}^j(c,k)^2\boldsymbol{\mu}^j(k)^2}{(\boldsymbol{\sigma}_{\mathbf{z}}^{j+1})^2(c)} + \frac{\left[(1-p)\times\boldsymbol{\mu}_{\mathbf{z}}^{j+1}(c)\right]^2}{p(\boldsymbol{\sigma}_{\mathbf{z}}^{j+1})^2(c)}\right\}$$

finding back (7).

$\square$

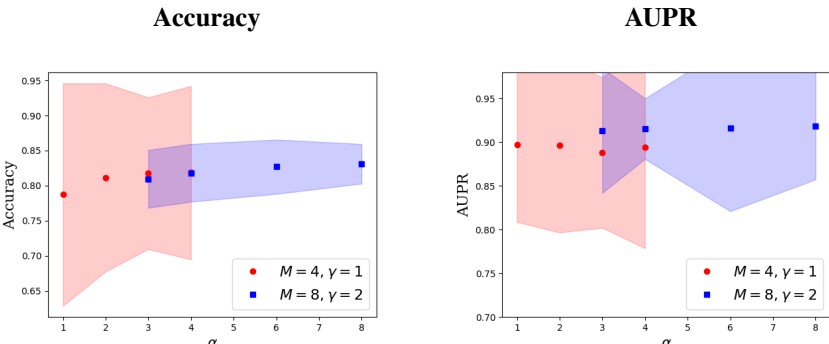

Figure 6: Accuracy and AUPR of Packed-Ensembles with ResNet-50 on CIFAR-100 depending on $\alpha$.

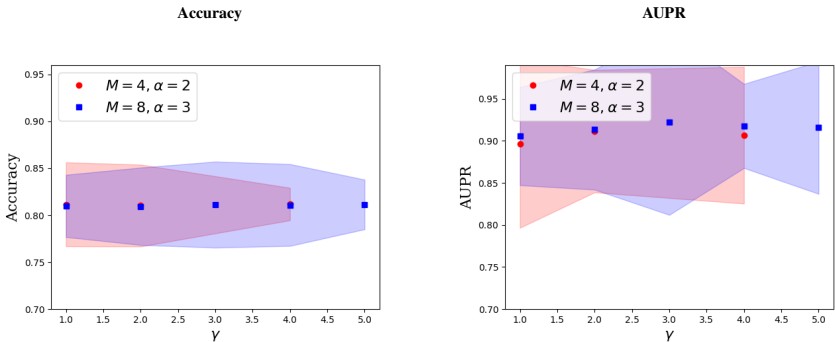

Figure 7: Accuracy and AUPR of Packed-Ensembles with ResNet-50 on CIFAR-100 depending on $\gamma$.

## D  ABLATION STUDY

Our algorithm mainly depends on three hyperparameters. M represents the number of subnetworks in the ensemble, $\alpha$ controls the power of representation of the DNN, and $\gamma$ is an extra parameter that controls the sparsity degree of the DNN. To evaluate the sensitivity of Packed-Ensembles to these parameters, we train 5 ResNet-50 on CIFAR-10 similarly to the protocol explained in section 4.1. Figures 6 and 7 show that the more we add subnetworks increasing $M$, the better the performance, in terms of accuracy and AUPR. We also note that the results are stable with $\gamma$. Moreover, the resulting accuracy tends to increase with $\alpha$ until it reaches a plateau. These statements are confirmed by the results in Table 5.

## E  DISCUSSION ABOUT OOD CRITERIA

Deep Ensembles (Lakshminarayanan et al., 2017) and Packed-Ensembles are ensembles of DNNs that can be used to quantify the uncertainty of the DNNs prediction. Similarly to Bayesian Neural Network, one can take the softmax outputs of posterior predictive distribution, which define the $\mathbf{MSP} = \max_{y_i}\{P(\mathbf{y}_i|\mathbf{x}, \mathcal{D})\}$. The MSP can also be used for classical DNN, yet we use the conditional likelihood instead of the posterior distribution in this case.

One can also use the Maximum Logit (ML) and the entropy of the posterior predictive distribution as uncertainty criteria, which is defined by $\mathbf{Ent.} = \mathcal{H}(P(\mathbf{y}_i|\mathbf{x}, \mathcal{D}))$ with $\mathcal{H}$ being the entropy function. Another metric is the mutual information between two random variables, which is defined by: $\mathrm{MI} = \mathcal{H}(P(\mathbf{y}_i|\mathbf{x}, \mathcal{D})) - \frac{1}{M}\sum_{m=0}^{M-1} \mathcal{H}(P(\mathbf{y}|\boldsymbol{\theta}_{\alpha,m}, \mathbf{x}))$. It represents a measure of the ensemble entropy, which is the entropy of the posterior minus the average entropy over predictions.

Table 5: **Performance (Acc / ECE / AUPR) of Packed-Ensembles for various $\alpha$ and $\gamma$ with ResNet-50 on CIFAR-100 and $M = 4$.**

| $\gamma$ \ $\alpha$ | 1 | 2 | 3 | 4 |
|---|---|---|---|---|
| 1 | 0.7872 / 0.0165 / 0.8969 | 0.8116 / 0.0203 / 0.8966 | 0.8187 / 0.0201 / 0.8825 | 0.8183 / 0.0230 / 0.8939 |
| 2 | 0.7857 / 0.0185 / 0.9024 | 0.8103 / 0.0295 / 0.9115 | 0.8186 / 0.0197 / 0.9127 | 0.8242 / 0.0190 / 0.9088 |
| 4 | / | 0.8119 / 0.0180 / 0.9066 | 0.8182 / 0.0236 / 0.9140 | 0.8225 / 0.0226 / 0.9229 |

Table 6: Comparison of the effect of the different uncertainty criteria for OOD on CIFAR-100 with different sets of parameters for Packed-Ensembles.

| Criterion | OOD eval | $\alpha = 2, \gamma = 1\, M = 4$ | $\alpha = 3, \gamma = 1\, M = 8$ | $\alpha = 4, \gamma = 2\, M = 8$ | $\alpha = 6, \gamma = 4\, M = 8$ | $\alpha = 8, \gamma = 1\, M = 16$ |
|---|---|---|---|---|---|---|
| **MSP** | AUPR ($\uparrow$) | $0.8952 \pm 0.0132$ | $0.9055 \pm 0.0034$ | $0.9153 \pm 0.0012$ | $0.9149 \pm 0.0071$ | $0.9141 \pm 0.0057$ |
| **ML** | AUPR ($\uparrow$) | $\mathbf{0.9183 \pm 0.0098}$ | $\mathbf{0.9175 \pm 0.0044}$ | $\mathbf{0.9285 \pm 0.0012}$ | $\mathbf{0.9265 \pm 0.0070}$ | $\mathbf{0.9268 \pm 0.0068}$ |
| **Ent.** | AUPR ($\uparrow$) | $0.9105 \pm 0.0138$ | $0.9152 \pm 0.0035$ | $0.9260 \pm 0.0016$ | $0.9237 \pm 0.0066$ | $0.9252 \pm 0.0060$ |
| MI | AUPR ($\uparrow$) | $0.8649 \pm 0.0061$ | $0.9139 \pm 0.0077$ | $0.9157 \pm 0.0072$ | $0.9196 \pm 0.0109$ | $0.9245 \pm 0.0091$ |
| **v** | AUPR ($\uparrow$) | $0.8404 \pm 0.0071$ | $0.8746 \pm 0.0056$ | $0.8827 \pm 0.0033$ | $0.8842 \pm 0.0102$ | $0.8931 \pm 0.0072$ |
| **MSP** | AUC ($\uparrow$) | $0.8056 \pm 0.0260$ | $0.8204 \pm 0.0101$ | $0.8408 \pm 0.0033$ | $0.8432 \pm 0.0134$ | $0.8387 \pm 0.0094$ |
| **ML** | AUC ($\uparrow$) | $\mathbf{0.8562 \pm 0.0194}$ | $\mathbf{0.8421 \pm 0.0115}$ | $\mathbf{0.8665 \pm 0.0027}$ | $\mathbf{0.8621 \pm 0.0144}$ | $\mathbf{0.8607 \pm 0.0114}$ |
| **Ent.** | AUC ($\uparrow$) | $0.8361 \pm 0.0271$ | $0.8427 \pm 0.0095$ | $0.8662 \pm 0.0027$ | $0.8617 \pm 0.0136$ | $0.8614 \pm 0.0096$ |
| MI | AUC ($\uparrow$) | $0.7711 \pm 0.0064$ | $0.8312 \pm 0.0135$ | $0.8402 \pm 0.0116$ | $0.8468 \pm 0.0163$ | $0.8513 \pm 0.0120$ |
| **v** | AUC ($\uparrow$) | $0.7305 \pm 0.0153$ | $0.7799 \pm 0.0129$ | $0.7943 \pm 0.0082$ | $0.7999 \pm 0.0166$ | $0.8092 \pm 0.0113$ |

The last metric, used in active learning, is the variation ratio (Beluch et al., 2018), which measures the dispersion of a nominal variable and is calculated as the proportion of predicted class labels that are not the modal class prediction. It is defined by: $\mathbf{v} = 1 - \frac{f_i}{M}$, where $f_i$ is the number of predictions falling into the modal class category.

In Table 6, the results for the different metrics are reported. We note that **ML** seems to be the best metric to detect OOD. This metric is followed by **Ent.** and then MI. Note that **v**, widely used in active learning, does not seem effective in detecting OOD samples. This shows us that it is essential to use a good criterion in addition to good ensembling.

## F  DISCUSSION ABOUT THE SOURCES OF STOCHASTICITY

As written in the introduction of the paper, diversity is essential to the success of ensembling, be it for its accuracy but also for calibration and OOD detection. Three primary sources can induce weight diversity, and therefore diversity in the function space, during the training. These sources are the initialization of the weights, the composition of the batches, and the use of non-deterministic backpropagation algorithms [4]. On Table 7, we measure the performance and diversity of Packed-Ensembles trained on CIFAR-100. The quantity of diversity is measured by the mutual information and is twofold: we compute the in-distribution mutual information (ID**MI**) on the test set of CIFAR-100 and the OOD mutual information (OOD**MI**) on SVHN. Concerning the performance, we compute the accuracy, ECE, and AUPR, which are proxies of the quality of this diversity. Results of Table 7 lead to several takeaways. First, they hint that there is no clear best set of trivial sources of stochasticity. Except for the first (and greyed) line, which corresponds to ensembling completely identical networks (the training being totally deterministic, which the null MI confirms), the results seem equivalent in diversity (via mutual information) and ID/OOD performance. Secondly, it shows that the use of non-deterministic algorithms can be sufficient to generate diversity. It was noted that this effect does not always happen depending on the selected architecture and the precision used (`float16`, or `float32`).

Given that there is no emerging best set of stochasticity, we use the faster non-deterministic back-propagation algorithms and different initializations to ensure enough stochasticity and for programming convenience.

---

[4]see https://docs.nvidia.com/deeplearning/cudnn/api/index.html

Table 7: **Comparison of the diversities and the performance wrt. the different sources of stochasticity on CIFAR-100. ND** corresponds to the use of Non-deterministic backpropagation algorithms, **DI** to different initializations, and **DB** to different compositions of the batches. A standard error (over five runs) is included in small font.

| Stochasticity | | | ResNet-18 | | | | |
|---|---|---|---|---|---|---|---|
| ND | DI | DB | Acc (↑) | ECE (↓) | AUPR (↑) | IDMI | OODMI |
| - | - | - | 71.70±0.06 | 0.0497±0.0013 | 87.32±0.91 | 0±0 | 0±0 |
| ✓ | - | - | 75.79±0.22 | 0.0365±0.0044 | 89.53±0.47 | 0.1945 | 0.4001 |
| - | ✓ | - | 76.20±0.04 | 0.0419±0.0006 | 89.54±0.39 | 0.2011 | 0.4391 |
| - | - | ✓ | 76.06±0.02 | 0.0434±0.0011 | 88.70±0.27 | 0.1987 | 0.4079 |
| ✓ | ✓ | - | 76.10±0.05 | 0.0424±0.0004 | 88.65±0.42 | 0.1995 | 0.4360 |
| ✓ | - | ✓ | 76.19±0.11 | 0.0433±0.0010 | 88.87±0.15 | 0.2032 | 0.4090 |
| - | ✓ | ✓ | 76.14±0.07 | 0.0437±0.0008 | 89.21±0.38 | 0.1943 | 0.4195 |
| ✓ | ✓ | ✓ | 76.29±0.07 | 0.0445±0.0006 | 89.00±0.54 | 0.1954 | 0.4060 |
| Stochasticity | | | ResNet-50 | | | | |
| - | - | - | 77.63±0.23 | 0.0825±0.0018 | 89.19±0.65 | 0±0 | 0±0 |
| ✓ | - | - | 80.94±0.10 | 0.0179±0.0010 | 90.23±0.62 | 0.1513 | 0.4022 |
| - | ✓ | - | 81.01±0.06 | 0.0202±0.0011 | 91.10±0.39 | 0.1524 | 0.4088 |
| - | - | ✓ | 80.87±0.10 | 0.0178±0.0010 | 90.80±0.30 | 0.1505 | 0.4115 |
| ✓ | ✓ | - | 81.16±0.10 | 0.0210±0.0008 | 91.69±0.56 | 0.1584 | 0.4135 |
| ✓ | - | ✓ | 81.14±0.07 | 0.0200±0.0007 | 90.41±0.39 | 0.1503 | 0.3897 |
| - | ✓ | ✓ | 81.10±0.05 | 0.0186±0.0016 | 90.85±0.29 | 0.1521 | 0.4034 |
| ✓ | ✓ | ✓ | 81.08±0.08 | 0.0198±0.0013 | 90.68±0.25 | 0.1534 | 0.4031 |

## G    DISCUSSION ABOUT THE SUBNETWORKS

The width and depth of deep neural networks are crucial research topics, and researchers strive to determine the best approaches for increasing the depth of DNNs, which can lead to improved accuracy. According to Nguyen et al. (2020), the width and depth of a DNN are connected with its capacity to learn block structures, which can improve accuracy. Therefore, the model's capacity may decrease if the width is divided.

Deep neural networks are heavily over-parameterized, as stated by the lottery ticket hypothesis (Frankle & Carbin, 2018). It suggests that up to 80% of neurons can be removed without significant loss of performance. The MIMO approach builds on this assumption by allowing multiple networks to be trained simultaneously, and neurons may be used by several subnetworks. In our work, however, we assign each neuron to a specific DNN in the ensemble, guaranteeing their independence. This way, the DNNs can learn independent representations. However, as in MIMO, we rely on the fact that not all neurons are helpful, so we split the width of the initial DNNs into a set of DNNs. Although the decomposition may seem crude, it facilitates better parallelization of Packed-Ensembles during training and inference.

To address the problem of not having sufficiently wide subnetworks, we added a hyperparameter - $\alpha$ - to increase the width of subnetworks. In Figure 8, we explore the impact of subnetwork width.

Our observations reveal that the accuracy of the DNN increases with the width while the AUPR remains relatively constant. This finding suggests that the $\alpha$ is paramount in maintaining a balance in the DNN's width. We also note that reducing the width of the DNN does not significantly impact its accuracy. Hence, our decision to split the width of the DNN to create multiple subnetworks is justified since the uncertainty quantification remains unaltered, and the accuracy is not significantly compromised.

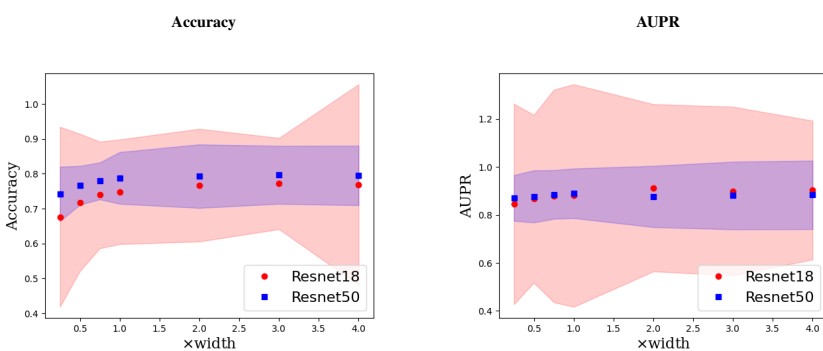

Figure 8: Accuracy and AUPR curves of ResNet-18 in red and ResNet-50 in blue on CIFAR-100 with different widths. When the width is equal to 1, it corresponds to the original ResNet; when the width is equal to $x$, the width of every layer is multiplied by $x$.

Table 8: **Comparison of training and inference times of different ensemble techniques** using torch1.12.1+cu113 on an RTX 3090. All ensembles have four subnetworks.

| models | f32 Precision | | f16 Precision | |
| | Training ↓ | Inference ↑ | Training ↓ | Inference ↑ |
| *ResNet-50* *CIFAR-100* | s/epoch | im/s | s/epoch | im/s |
| --- | --- | --- | --- | --- |
| Single Model | 37.06 | 3709 | 22.42 | 5718 |
| Packed-Ensembles-(2,4,1) | 179.50 | 1381 | 51.20 | 3406 |
| Packed-Ensembles-(2,4,2) | 175.10 | 1501 | 52.11 | 3440 |
| Deep Ensembles | 145.30 | 1001 | 84.86 | 1609 |
| MIMO | 37.90 | 3574 | 24.44 | 5649 |
| BatchEnsemble | 58.78 | 1809 | 53.97 | 1916 |

In addition, $\alpha$ provides an additional degree of freedom to our ensemble, enabling it to enhance its accuracy. This is a significant advantage, as it allows us to further balance the performance of the ensemble, which can lead to more accurate predictions - and the number of parameters linked to its computational cost.

## H    DISCUSSION ABOUT THE TRAINING VELOCITY

Our experiments show that grouped convolutions are not as fast as they could theoretically be, and confirm the statements made by many PyTorch and TensorFlow users [5]. Following the idea that grouped convolutions are bandwidth-bound, we advise readers to leverage Native Automatic Mixed Precision (AMP) and cuDNN benchmark flags when training a Packed-Ensembles to reduce the bandwidth bottleneck compared to the baseline. AMP also divides the VRAM usage by two while yielding equally good results. Future improvements of PyTorch grouped convolutions should help Packed-Ensembles develop its full potential, increasing its current assets. We note in Table 8 that using `float16`, Packed-Ensembles is only $1.6\times$ slower than the single model during inference. Furthermore, Packed-Ensembles is only $2.3\times$ slower during training than the single model, making it an efficient model capable of training four models in half the time of a Deep Ensembles.

## I    DISTRIBUTION SHIFT

In this section, we evaluate the robustness of Packed-Ensembles under dataset shift. We use models trained on CIFAR-100 (Krizhevsky, 2009) and shift the data using corruptions and perturbations proposed by (Hendrycks & Dietterich, 2019) to produce CIFAR-100-C. There are five levels of perturbations called "severity", from one, the weakest, to five, the strongest. In real-world scenarios,

---

[5]For instance https://github.com/pytorch/pytorch/issues/75747

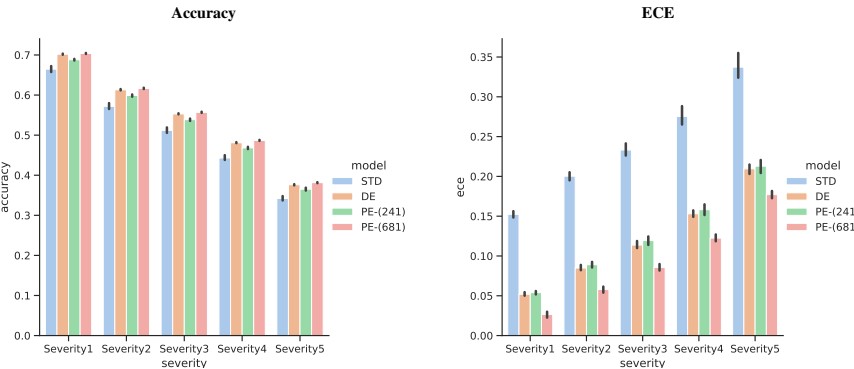

Figure 9: Accuracy and Calibration under distributional shift. Comparison of the accuracy and ECE under all types of corruptions on (a) CIFAR-100-C (Hendrycks & Dietterich, 2019) with different levels of severity.

distributional shift is crucial, as explained by (Ovadia et al., 2019), and it is critical to study how much a model prediction shifts from the original training data distribution. Thanks to Figure 9, we notice that Packed-Ensembles achieves the highest accuracy and lowest ECE under distributional shift, leading to a method robust against this uncertainty.

## J    STABILIZATION OF THE PERFORMANCE

We perform five times each training task on CIFAR-10 and CIFAR-100 to estimate a better value and be able to compute the variance. Let us first note that the standard deviation for the single DNN on CIFAR-100 with a ResNet-50 architecture amounts to $0.68\%$. Ensemble strategies shrink the standard variation to $0.43\%$ for Deep Ensembles and $0.19\%$ for Packed-Ensembles. Thus it seems that Packed-Ensembles makes DNN predictions more stable in addition to improving accuracy and uncertainty quantification. This result is interesting as it appears to contradict Neal et al. (2019), who claim that wider DNNs have a smaller variance. This stability might come from the ensembling.

## K    ON THE EQUIVALENCE BETWEEN SEQUENTIAL TRAINING AND PACKED-ENSEMBLES

The sequential training of Deep Ensembles differs significantly from the training procedure of Packed-Ensembles. The main differences lie in the subnetworks' batch composition and the best models' selection.

Concerning Packed-Ensembles, the batches are strictly the same for all subnetworks, thus removing one source of stochasticity compared to sequential learning. Yet, in practice, we show empirically that random initialization and stochastic algorithms are sufficient to get diverse subnetworks (see Appendix F for more details).

For the selection of models, Packed-Ensembles considers subnetworks as a whole (i.e., maximize the ensemble accuracy on the validation set) and therefore selects the best ensemble at a given epoch. On the other hand, sequential training selects the best networks individually, possibly on different epochs, which does not guarantee that the best ensemble is selected but ensures the optimality of subnetworks over the epochs.

## L    USING GROUPS IS NOT SUFFICIENT TO EQUAL PACKED-ENSEMBLES

To make sure that the use of groups cannot simply explain our results, we compare Packed-Ensembles to a single ResNeXt-50 (32×4d) (Xie et al., 2017) in Table 9. ResNeXt-50 is fairly equivalent to our method but does not propagate groups, only used in the middle layer of each

Table 9: **Comparison between the results obtained with Packed-Ensembles and a similar ResNeXt-50**. The dataset is CIFAR-10.

| Network | Acc | NLL | ECE | AUPR | AUC | FPR95 | Params (M) |
|---------|-----|-----|-----|------|-----|-------|------------|
| PE ResNet-50 | **96.0** | **0.1367** | **0.0087** | **97.1** | **94.9** | **14.5** | 23.6 |
| ResNeXt-50 | 90.4 | 0.4604 | 0.0709 | 90.4 | 82.5 | 63.4 | **23.0** |

Table 10: **Comparison of the efficiency of the networks trained on ImageNet (Deng et al., 2009)**. All ensembles have $M = 4$ subnetworks and $\gamma = 1$. *Mult-Adds* corresponds to the inference cost, i.e., the number of Giga multiply-add operations for a forward pass which is estimated with Torchinfo (2022).

| Method | Net | **Params** (M) $\downarrow$ | **Mult-Adds** (G) $\downarrow$ |
|--------|-----|------------|--------------|
| Single Model | R50 | 25.6 | 4.09 |
| BatchEnsemble | R50 | 25.7 | 16.36 |
| MIMO | R50 | 31.7 | 4.45 |
| Masksembles | R50 | 25.7 | 16.36 |
| Packed-Ensembles ($\alpha = 3$) | R50 | 59.1 | 9.29 |
| Deep Ensembles | R50 | 102.4 | 16.36 |
| Single Model | R50x4 | 383.6 | 70.0 |
| BatchEnsemble | R50x4 | 384.4 | 256.0 |
| MIMO | R50x4 | 408.3 | 65.4 |
| Masksembles | R50x4 | 384.0 | 256.0 |
| Packed-Ensembles ($\alpha = 2$) | R50x4 | 392.0 | 64.47 |
| Deep Ensembles | R50x4 | 1534.4 | 280.0 |

block, which are therefore not independent. We keep the same training optimization procedures and data-augmentation strategies detailed in Appendix B.

## M  EFFICIENCY OF THE NETWORKS TRAINED ON IMAGENET

Table 10 provides the efficiency of the networks trained on ImageNet-1k (see section 4.1.3), in number of parameters and multiply-additions. PE-(3, 4, 1) was preferred to PE-(3, 4, 2) for ResNet50 to improve the representation capacity of the subnetworks.

## N  REGRESSION

To generalize our work, we propose to study regression tasks. We replicate the setting developed by Hernández-Lobato & Adams (2015), Gal & Ghahramani (2016), and Lakshminarayanan et al. (2017).

For the training in the one-dimensional regression setting, we minimize the gaussian NLL (20) using networks with two outputs neurons which estimate the parameters of a heteroscedastic gaussian distribution (Nix & Weigend, 1994; Kendall & Gal, 2017). One output corresponds to the mean of the predicted gaussian distribution, and the softplus applied on the second is its variance. The ensemble's mean $\bar{\mu}_\theta(\mathbf{x}_i)$ is computed using the empirical mean over the estimators and the variance using the formula of a mixture $\bar{\sigma}_\theta(\mathbf{x}_i)^2 = M^{-1} \sum_m \left( \sigma_{\theta_m}(\mathbf{x}_i)^2 + \mu_{\theta_m}(\mathbf{x}_i)^2 \right) - \bar{\mu}_\theta(\mathbf{x}_i)$ (Lakshminarayanan et al., 2017).

$$\mathcal{L}\left(\mu_{\theta_m}(\mathbf{x}_i), \sigma_{\theta_m}(\mathbf{x}_i)^2, y_i\right) = \frac{(y_i - \mu_{\theta_m}(\mathbf{x}_i))^2}{2\sigma_{\theta_m}(\mathbf{x}_i)^2} + \frac{1}{2}\log\sigma_{\theta_m}(\mathbf{x}_i)^2 + \frac{1}{2}\log 2\pi \qquad (20)$$

We compare Packed-Ensembles-(2, 3, 1) and Deep Ensembles on the UCI datasets in Table 11. The subnetworks of these methods are based on multi-layer perceptrons with a single hidden layer, containing 400 neurons for the more extensive Protein dataset and 200 for the others, and a ReLU non-linearity. The results show that Packed-Ensembles and Deep Ensembles provide equivalent results on most datasets.

Table 11: **Comparison of the results obtained with Packed-Ensembles and Deep Ensembles on regression tasks.**

| Datasets | RMSE | | NLL | |
|---|---|---|---|---|
| | Packed-Ensembles | Deep Ensembles | Packed-Ensembles | Deep Ensembles |
| Boston housing | **2.218 ± 0.099** | **2.219 ± 0.098** | **2.028 ± 0.034** | 2.047 ± 0.028 |
| Concrete | **5.092 ± 0.225** | 5.167 ± 0.234 | **2.854 ± 0.028** | 2.885 ± 0.032 |
| Energy | **1.675 ± 0.085** | 1.712 ± 0.067 | **1.543 ± 0.072** | **1.553 ± 0.060** |
| Kin8nm | **0.058 ± 0.003** | **0.058 ± 0.003** | -1.442 ± 0.010 | **-1.452 ± 0.010** |
| Naval Propulsion Plant | **0.002 ± 0.000** | **0.002 ± 0.000** | **-4.835 ± 0.066** | -4.833 ± 0.097 |
| Power Plant | 3.127 ± 0.018 | **3.097 ± 0.020** | 2.607 ± 0.007 | **2.600 ± 0.007** |
| Protein | 3.476 ± 0.030 | **3.412 ± 0.017** | 2.472 ± 0.033 | **2.442 ± 0.015** |
| Wine | **0.482 ± 0.006** | **0.483 ± 0.006** | 0.622 ± 0.014 | **0.611 ± 0.013** |
| Yacht | **1.949 ± 0.215** | 2.511 ± 0.283 | **2.023 ± 0.075** | **2.023 ± 0.074** |

