# OpenReview forum: "Packed Ensembles for efficient uncertainty estimation"
_ICLR.cc/2023/Conference — ICLR 2023 notable top 25%_

### Official Review · Reviewer_x31y · 2022-10-25

**Confidence:** 4
**Clarity, Quality, Novelty And Reproducibility:** The paper is well written and easy to…
**Correctness:** 3
**Technical Novelty And Significance:** 3
**Empirical Novelty And Significance:** 3
**Recommendation:** 6

**Strength And Weaknesses:**

Strength:
1. The proposed group convolution based ensemble has comparable or better performance with a full ensemble, using less parameters and computation.
2. The method is easy to implement with standard high-level neural network programming library.
3. The results on ImageNet is persuasive. Although not shown in the comparisons, it outperforms the Rank-1 BNN which uses the BatchEnsemble as backbone, in terms of accuracy and other important metrics.

Weaknesses:
1. There are some missing results on ImageNet about efficiency. Why is that?


**Summary Of The Paper:**

This paper proposes a simple method for the neural network ensemble by integrating the ensemble into one neural network with the group convolution operation. Empirical evaluations show a strong performance and space efficiency of the proposed method.

**Summary Of The Review:**

In general, this paper proposes a simple and useful ensemble technique.

---

> ### Author Response · Authors · 2022-11-15
> **Author response to Reviewer x31y**
>
> Dear Reviewer,
>
> We thank you for the encouragement and the kind feedback.
>
> ### Missing results on ImageNet:
> The efficiency results were not included in the original paper due to space constraints. Please find the values below (including new baselines):
>
> |                             |       | Params (M) | Mult-Adds |
> |-----------------------------|:----:|:----:|:----:|
> | Single Model                | R50   |      25.6       |    4.09  |
> | BatchEnsemble               | R50   |   25.7    |    16.36   |
> | MIMO                        | R50   |    31.7    |    4.45   |
> | Masksembles                 | R50   |    25.7     |    16.36  |
> | Packed-Ensembles ($\alpha = 3 $) | R50   |   59.1   |    9.29  |
> | Deep Ensembles              | R50   |    102.4   |    16.36   |
> | Single Model                | R50×4 |     383.6        |  70.0  |
> | BatchEnsemble               | R50×4 |   384.4    |   256.0 |
> | MIMO                        | R50×4 |   408.3   |   65.4  |
> | Masksembles                 | R50×4 |   384.0    |   256.0   |
> | Packed-Ensembles ($\alpha = 2$)  | R50×4 |   392.0  |   64.47   |
> | Deep Ensembles              | R50×4 |  1534.4   |    280.0 |
>
> We added this information in appendix M.
>
> ### General assessment of the paper
> Please note that we also added new baselines on ImageNet.
>
> Authors

---

> > ### Comment · Reviewer_x31y · 2022-12-08
> > **Thanks for the additional results.**
> >
> > Thanks for addressing my concerns. I would keep the positive score with a higher confidence.

---

### Official Review · Reviewer_FMDg · 2022-10-26

**Confidence:** 4
**Correctness:** 4
**Technical Novelty And Significance:** 3
**Empirical Novelty And Significance:** 3
**Recommendation:** 8

**Clarity, Quality, Novelty And Reproducibility:**

Quality: This paper is of good quality. The results are relevant to the community, the method well motivated and experimentally validated.
Clarity: the paper is clearly exposed, its position within the greater literature is well discussed.
Originality: This reuses many key ideas from previous work, with a novel perspective.

**Strength And Weaknesses:**

# Strengths
- This paper proposes a simple technique to improve the inference and memory cost of deep ensembles.
- Experimental results confirm the value of this method compared to commonly used "efficient ensemble" baselines such as MIMO and BatchEnsemble on Cifar datasets, across a variety of different metrics *and* architectures: the proposed PE model comes close, and sometime outperforms, the performance of deep ensembles on accuracy and uncertainty metrics, while drastically cutting down on the number of parameters and required operations.
- The authors provide an extensive amount of ablation experiments to gauge the importance of the $\alpha$ and $\gamma$ parameters.

# Weaknesses
- It would be interesting to evaluate the other baselines on ImageNet tasks. I realize this can be an investment, but doing so would significantly improve the paper. Some results (for $M=4$ presumably, given reported parameter counts) are publicly available at the link referenced in footnote 2.
- Although this is not necessary given the scope of this work, the authors may be interested in looking into the Mixtures of Experts literature, as decisions in data routing and partitioning may be also relevant.

**Questions**
- I'm curious about the position of MIMO in Figure 1: it looks like MIMO (4) has the same number of parameters as a single model, but much slower inference. Could you explain what causes this?

**Nitpicks**
- Equation (1) doesn't make use of the activation function $\phi$.

**Summary Of The Paper:**

The authors introduce PackedEnsembles (PE) to improve the efficiency of deep ensembles. Although deep ensembles achieve SOTA results on a variety of benchmarks, this comes at significant inference-time and memory costs, since the same model architecture is repeated $M$ times for ensembles of size $M$. In this sense, this paper belongs to the same line of research as MIMO and BatchEnsembles (and extensions thereof), as well as mixtures of experts (MoEs) to a lesser degree.

PackedEnsembles rely fundamentally upon the notion of grouped convolutions, first introduced in (Krizhevsky et al., 2012) for hardware constraint reasons, and used in more modern architectures such as ResNext. Grouped convolutions essentially partition the filters at a given convolution layer into $M$ groups, and each output channel is obtained by the filters in of such groups (as opposed to all filters).

As this reduces significantly the capacity of each individual "ensemble member", the authors introduce a parameter $\alpha$, which multiplies the width of the original model (resulting in ensemble members of size rescaled roughly by $\alpha / M$). Furthermore, each individual ensemble member may itself make use of grouped convolutions, parameterized by the number of groups $\gamma$.

**Summary Of The Review:**

This paper proposes a straightforward, well-motivated, and experimentally validated technique to approach the performance of deep ensembles while significantly improving upon the complexity of deep ensembles. The results are validated across several ResNet-based architectures and standard image benchmarks. This paper could be improved by including more baselines for the ImageNet experiments, as the Cifar datasets are significantly easier; however, achieving this can be costly.

---

> ### Author Response · Authors · 2022-11-15
> **Response to reviewer FMDg**
>
> Dear Reviewer,
>
> Thanks a lot for your detailed comments and valuable feedback. We appreciate your review and plan to make the following changes in the paper to better address your concerns:
>
> ### Additional ImageNet baselines
>
> Following the reviewer's suggestion we thought about additional baselines to be included on the ImageNet experiments (Table 2), namely: BatchEnsemble(4), MaskEnsemble(4) and MIMO(4). For fair comparison we test them with ResNet-50 and with ResNet-50x4 backbones.
> We report here some of the results for experiments that have just concluded so far and we recall results of Packed-Ensembles. In the short time we had, we could not make MaskEnsemble work on ResNet-50x4 and share only ResNet50 results.
> We report our results below:
> | Method        | Net   | Acc  | ECE    | AUPR (T) | AUC (T) | FPR95 (T) | AUPR (IO) | AUC (IO) | FPR95 (IO) | rAcc | rNLL | rECE |
> |---------------|:----:|:----:|:----:|:----:|:----:|:----:|:----:|:----:|:----:|:----:|:----:|:----:|
> | BatchEnsemble | R50   | 75.9 | 0.0348 |   20.2 | 81.6    | 66.5      |   4.0 | 55.2     | 82.3       | 21.0 | 6.148 | 0.1649 |
> | MIMO          | R50   | 77.6 | 0.1465 |     18.4 | 81.6    | 66.8      |       3.7 | 52.2     | 90.6       |23.4 | 5.115 | 0.0585 |
> | Masksembles   | R50   | 73.6 | 0.2093 |     13.6 | 79.7    | 68.3      | 3.31      | 47.7     | 87.7       | 21.2  | 5.139 | 0.0107 |
> | Packed-Ensembles $\alpha=3$   | R50   | 77.9 | 0.1796 |   35.1 |  88.2    | 43.7      | 9.9      | 68.4    | 80.9       | 23.8 | 4.978 | 0.0221 |
> | MIMO          | R50×4 | 80.3 | 0.0150 | 19.3     | 82.5    | 66.1      | 4.9       | 60.7     | 86.4       |25.8 |5.278 | 0.1886 |
> | Packed-Ensembles $\alpha=2$   | R50×4   | 82.1 | 0.1034 |   34.6 |  88.1    | 50.3      | 9.6      | 69.9    | 79.2    | 26.6 | 4.848 | 0.0750 |
>
> ### MIMO inference speed
>
> Following the reviewer's remark we double checked our implementation and realized that indeed the reported times for MIMO were not correct. In fact due to the specific batch repetition operation of MIMO, the reported runtimes were a bit off. We recomputed inference speed and obtained for MIMO a throughput of 5600 images per second which, unsurprisingly, is close to the speed of the single model. We took advantage of your remark to double-check all the other inference times, which were correct.  We modified Figure 1 accordingly.
>
> ### Typos Concerning $\phi$
>
> We modified the paper and fixed its explanation.
>
> ### Advice on MoEs
>
> Thank you for bringing up Mixtures of Experts techniques to our attention. This indeed an interesting avenue of research for extending this work towards Transformer architectures. We could indeed exploit the Mixture of Experts formalism and instead of the routing mechanism (by soft or hard gating) we could propagate the features through all heads at once. At this point with a common backbone, the architecture would be similar with the one of TreeNet (a common backbone and multiple heads) and some care would be needed to make sure the experts are learning diverse representations.
> In order to obtain multiple subnetworks for the main Transformer backbone, we could potentially repurpose the recently proposed parallel vision transformer idea by Touvron et al. **[6]**.  Parallel vision transformer produces an
> architecture with the same number of parameters and compute, while being wider and shallower. This design allows for more parallel processing so we could imagine repurposing them in a similar manner we did with group convolutions.
> We leave this for exploration for future works.
>
> Authors
>
>
> ### Bibliography
> **[5]** Ross Wightman, Hugo Touvron, and Herve Jegou. Resnet strikes back: An improved training procedure in timm. In NeurIPS 2021 - Workshop ImageNet PPF, 2021
>
> **[6]** Touvron, Hugo et al. ”Three things everyone should know about Vision Transformers”, ECCV 2022

---

> > ### Comment · Reviewer_FMDg · 2022-12-06
> > **Rebuttal acknowledgement**
> >
> > Thank you for your changes! I'm glad to see the updated inference time for MIMO, and the final updated results on ImageNet (table 2 in the paper) are also convincing. Given the importance given to diversity in this paper (e.g., called out by name in the abstract), the authors may want to include a subset of Table 7 in the main paper.

---

### Official Review · Reviewer_GYMm · 2022-11-05

**Confidence:** 4
**Correctness:** 3
**Technical Novelty And Significance:** 3
**Empirical Novelty And Significance:** 2
**Recommendation:** 6

**Clarity, Quality, Novelty And Reproducibility:**

The clarity of this paper is good. As previously mentioned, the novelty and correspondingly the quality, is limited. Reproducibility is high.

**Strength And Weaknesses:**

Strength:

Simple and effective implementation with extensive comparisons and discussions. Overall a well written paper.

Weakness:

The introduction of group convolution into DE might be too incremental for ICLR. Also, it seems like this idea has been entertained before https://arxiv.org/abs/2007.00649.

I am not sure that I totally get the conclusions in Table 7. All the performance discrepancies are very marginal (maybe some hypothesis testing is needed? But 5 is too small a sample size to draw conclusions), and I don't think better ECE, NLL etc. leads to better diversity. In fact, the fuction space diversity can be easily checked by the correlations among predictions of different ensemble members. Maybe the authors wanted to add experiment here, similar to what the MIMO work did.

One downside of group conv is the scalability, i.e., with more ensemble members, the performance of group convolutions might drop. DE does not suffer from the same issue, and it would be interesting to test the performance of PE vs DE with an increasing number of ensemble members.

**Summary Of The Paper:**

This paper proposes an improved version of Deep Ensembled, i.e., Packed-Ensembles (PE), which cleverly leverages grouped convolutions to parallelize the ensemble into a single backbone to speed up the inference and training (eliminating the need to train multiple nns). The authors also extensively demonstrated the advantages of PE over the original DE.

**Summary Of The Review:**

In summary, I tend to accept this paper as it is a simple and effective approach to improve the performance of DE, but I would like to see a more thorough experiment section to examine the pro and con of PE. Thus, my recommendation is marginal accept.

---

> ### Author Response · Authors · 2022-11-15
> **Response to Reviewer GYMm**
>
> Dear Reviewer,
>
> We thank you for your thorough review.
>
> ### Group Ensemble Networks
>
> Our understanding is that in that work, the authors leverage grouped convolutions in order to equip a network with multiple prediction heads issued from a shared backbone. From this perspective, Group Ensemble seems close to a TreeNet **[1]** architecture that we mention in the original submission. We argue that our work comes with some substantial differences:
> - First, we leverage grouped convolutions to parallelize completely independent subnetworks (in a Deep Ensembles fashion). This brings the advantage of having more diversity in the ensemble predictions (as they are trained individually) but could reduce the capacity of the individual subnetworks
> - To address this issue, we emphasize our second difference, which is the use of the pair of parameters ($\alpha, \gamma$) to better control the trade-off between sparsity and representation capacity of the individual subnetworks. This further allows us to convert various architectures into different configurations of Packed-Ensembles according to the difficulty of the task to be solved.
> - In addition, in terms of scope, Group Ensemble focuses on accuracy, while we aim to devise a strategy to leverage ensembles towards improving not just accuracy but also the uncertainty, calibration, and diversity of predictions, i.e., the reliability of the network. We studied how these new layers can approximate an ensemble and added various studies to better understand this new design.
>
> ### Conclusions from Table 7
>
> In Table 7, we conduct an analysis of the impact of different sources of stochasticity (non-deterministic backpropagation, different initializations, different compositions of batches) for Ensembles on ResNet-18 and ResNet-50. For each configuration, we report average performance over five random seeds. We agree with the reviewer that the differences in performances are marginal. Our initial aim with this table was to emphasize that there does not seem to be a set of obvious sources of stochasticity that stands out in achieving best results.
> The only significant difference appears with ensembles with no sources of stochasticity (which amount to the mean of identical estimators) and which are displayed on the top line of Table 7. Concerning the assessment of the diversity of the PE networks, we proposed using Mutual Information to measure the diversity of this ensemble. The Mutual Information is equivalent to the generalized Jensen-Shannon divergence, which is a symmetrized version of the KL divergence used by the authors of MIMO **[2, 3]**.
>
> ### Scalability of PE
>
> We acknowledge the shortcomings of group convolutions concerning scalability for a larger ensemble. As described in section 3.2, for a fixed $\alpha$ parameter the number of parameters would linearly decrease as the number of members grows, resulting in the shrinkage of the subnetworks’ capacity and poor ensemble performance. However, we argue that one could make use of the $\alpha$ parameter to expand the network width thus allowing more ensemble members without a large performance drop.
> We conduct several sensitivity studies in this line in the Appendix:
> - In Figure 6 in Appendix D we compare PE with an ensemble of 4 and respectively 8 networks and for different values of the $\alpha$ parameter ( $\in [1,8]$) and show that bigger ensembles bring  a slight boost in AUPR but preserve accuracy.
> - We conduct a similar study with the impact of the $\gamma$ parameter in Figure 7.
> - In Table 6 (Appendix E) we evaluate OOD performance for PE with ResNet-50 composed of 4 to 16 subnetworks. The performance improves with bigger ensembles, but the best trade-off seems to be for M=4 or M=8 subnetworks which is backed by observations from **[4]**.
>
> We will shortly provide another version of our manuscript and to complete our answers and propose other improvements.
>
> Authors
>
> ### Bibliography
>
> **[1]** Lee, Stefan, et al. "Why M Heads are Better than One: Training a Diverse Ensemble of Deep Networks." arXiv preprint arXiv:1511.06314, 2015
>
> **[2]** Goldberger, Jacob, and Yaniv Opochinsky. "Information-bottleneck Based on the Jensen-shannon Divergence with Applications to Pairwise Clustering.", ICASSP, 2019
>
> **[3]** Grosse, Ivo, et al. "Analysis of symbolic sequences using the Jensen-Shannon divergence." Physical Review, 2002
>
> **[4]** Ashukha, Arsenii, et al. “Pitfalls of In-Domain Uncertainty Estimation and Ensembling in Deep Learning”, ICLR 2020

---

> > ### Author Response · Authors · 2022-11-19
> > **More details on the response to reviewer GYMm**
> >
> > ### On the conclusions of Table 7
> > Please find here the complete table including standard errors. We deliberately choose not to provide statistical testing considering the sizes of the samples. We also added the out-of-distribution Mutual Information to quantify and compare the OOD diversities across the different sources of stochasticity. Please note that the first line (below the architecture) corresponds to the case with no stochasticity and therefore amounts to a Deep Ensemble of exactly identical networks, which is therefore equivalent to a classical single network.
> >
> > ||  |  || ResNet-18 |  |  |  |
> > | :---: | :---: | :---: | :---: | :---: | :---: | :---: | :---: |
> > | ND | DI | DB | Acc $(\uparrow)$ | ECE $(\downarrow)$ | AUPR $(\uparrow)$ | IDMI | OODMI |
> > | $-$ | $-$ | $-$ | $71.70 \pm 0.06$ | $0.0497 \pm 0.0013$ | $87.32 \pm 0.91$ | $0 \pm 0$ | $0 \pm 0$ |
> > | $\checkmark$ | $-$ | $-$ | $75.79 \pm 0.22$ | $0.0365 \pm 0.0044$ | $89.53 \pm 0.47$ | $0.1945$ | $0.4001$ |
> > | $-$ | $\checkmark$ | $-$ | $76.20 \pm 0.04$ | $0.0419 \pm 0.0006$ | $89.54 \pm 0.39$ | $0.2011$ | $0.4391$ |
> > | $-$ | $-$ | $\checkmark$ | $76.06 \pm 0.02$ | $0.0434 \pm 0.0011$ | $88.70 \pm 0.27$ | $0.1987$ | $0.4079$ |
> > | $\checkmark$ | $\checkmark$ | $-$ | $76.10 \pm 0.05$ | $0.0424 \pm 0.0004$ | $88.65 \pm 0.42$ | $0.1995$ | $0.4360$ |
> > | $\checkmark$ | $-$ | $\checkmark$ | $76.19 \pm 0.11$ | $0.0433 \pm 0.0010$ | $88.87 \pm 0.15$ | $0.2032$ | $0.4090$ |
> > | $-$ | $\checkmark$ | $\checkmark$ | $76.14 \pm 0.07$ | $0.0437 \pm 0.0008$ | $89.21 \pm 0.38$ | $0.1943$ | $0.4195$ |
> > | $\checkmark$ | $\checkmark$ | $\checkmark$ | $76.29 \pm 0.07$ | $0.0445 \pm 0.0006$ | $89.00 \pm 0.54$ | $0.1954$ | $0.4060$ |
> > | | |  |  | **ResNet-50** |  |  |  |  |
> > | $-$ | $-$ | $-$ | $77.63 \pm 0.23$ | $0.0825 \pm 0.0018$ | $89.19 \pm 0.65$ | $0 \pm 0$ | $0 \pm 0$ |
> > | $\checkmark$ | $-$ | $-$ | $80.94 \pm 0.10$ | $0.0179 \pm 0.0010$ | $90.23 \pm 0.62$ | $0.1513$ | $0.4022$ |
> > | $-$ | $\checkmark$ | $-$ | $81.01 \pm 0.06$ | $0.0202 \pm 0.0011$ | $91.10 \pm 0.39$ | $0.1524$ | $0.4088$ |
> > | $-$ | $-$ | $\checkmark$ | $80.87 \pm 0.10$ | $0.0178 \pm 0.0010$ | $90.80 \pm 0.30$ | $0.1505$ | $0.4115$ |
> > | $\checkmark$ | $\checkmark$ | $-$ | $81.16 \pm 0.10$ | $0.0210 \pm 0.0008$ | $91.69 \pm 0.56$ | $0.1584$ | $0.4135$ |
> > | $\checkmark$ | $-$ | $\checkmark$ | $81.14 \pm 0.07$ | $0.0200 \pm 0.0007$ | $90.41 \pm 0.39$ | $0.1503$ | $0.3897$ |
> > | $-$ | $\checkmark$ | $\checkmark$ | $81.10 \pm 0.05$ | $0.0186 \pm 0.0016$ | $90.85 \pm 0.29$ | $0.1521$ | $0.4034$ |
> > | $\checkmark$ | $\checkmark$ | $\checkmark$ | $81.08 \pm 0.08$ | $0.0198 \pm 0.0013$ | $90.68 \pm 0.25$ | $0.1534$ | $0.4031$ |
> >
> > This table tends to confirm that we cannot expect one set of sources of stochasticity to provide better results than another (excepted for a training without source of stochasticity, which is not a real ensemble).

---

### Author Response · Authors · 2022-11-15
**General response to reviews**

We thank the reviewers for their thoughtful and helpful remarks. We appreciate your constructive comments, which we answered individually. For our revised manuscript, we make the following significant changes:
- We improve the quality of the paper by adding baselines.
- We provide the associated performance on CIFAR and ImageNet.
- We provide the efficiency results on ImageNet.

All the changes in the revision are highlighted in red text.

We will very shortly provide another version of this manuscript including the complete results on ImageNet.

We would be happy to swiftly respond and clarify any remaining question,

Authors

---

> ### Author Response · Authors · 2022-11-19
> **Update of the general response at the end of the first discussion stage**
>
> We thank the reviewers for their patience. At the end of the rebuttal phase, we upload a revised version of the manuscript integrating the feedback from the reviewers. All changes are marked with red color text.
> In addition to the previous update a few days ago, we made the following significant changes:
> - We have added a reference to GroupEnsemble
> - We report the standard error and out-of-distribution Mutual Information  in Table 7
>
> We intend to enrich the comparison on ImageNet with further baselines such as MaskEnsembles and BatchEnsembles with ResNet 50x4 backbone. Our computing cluster was particularly loaded these few weeks and we were unable to run all experiments that we had foreseen and launched for the rebuttal period. We apologize for this.

---

### Decision · Program_Chairs · 2023-01-20

**Decision:**

Accept: notable-top-25%

**Justification For Why Not Higher Score:**

There are some limitations, such as how big the number of ensemble members can be, and it's not clear yet how much the idea translates to other architectures and tasks.

**Justification For Why Not Lower Score:**

The proposal is simple (a big plus in my mind) and the empirical results are quite convincing to reviewers. Further empirical comparison to related work is added during the rebuttal that swayed less sure reviewers. I think it has especially neat connections to ensembles as a single model that are implicitly split into subnetworks like MIMO.

**Metareview: Summary, Strengths And Weaknesses:**

Ensembling strategies have been powerful for a wide variety of uncertainty and out-of-distribution generalization problems. However, there's been growing interest in efficient ensembling strategies building on ideas such as BatchEnsemble and MIMO. This work proposes an alternative strategy to enable efficient ensembles called Packed Ensembles. The idea is based on applying grouped convolutions in order to parallelize training and inference speeds, and the work is designed to work under a relatively similar memory budget as the original neural network.

The proposal is simple (a big plus in my mind) and the empirical results are quite convincing to reviewers. Further empirical comparison to related work is added during the rebuttal that swayed less sure reviewers. I think it has especially neat connections to ensembles as a single model that are implicitly split into subnetworks like MIMO.

Overall, everyone seems quite positive and leans towards accept. I agree with this consensus.

**Note From Pc:**

if the above contains the word "oral" or "spotlight" please see: "oral" presentation means -> notable-top-5% and "spotlight" means -> notable-top-25%. As stated in our emails, we are disassociating presentation type from AC recommendations